# Structure of HIV-1 gp41 with its membrane anchors targeted by neutralizing antibodies

**Christophe Caillat[1†], Delphine Guilligay[1†], Johana Torralba[2], Nikolas Friedrich[3], Jose L Nieva[2], Alexandra Trkola[3], Christophe J Chipot[4,5,6], François L Dehez[4,5], Winfried Weissenhorn[1]***

[1]Univ. Grenoble Alpes, CEA, CNRS, Institut de Biologie Structurale (IBS), Grenoble, France; [2]Instituto Biofisika (CSIC, UPV/EHU) and Department of Biochemistry and Molecular Biology, University of the Basque Country (UPV/EHU), Bilbao, Spain; [3]Institute of Medical Virology, University of Zurich, Zurich, Switzerland; [4]Laboratoire de Physique et Chimie Théoriques (LPCT), University of Lorraine, Vandoeuvre-lès-Nancy, France; [5]Laboratoire International Associé, CNRS and University of Illinois at Urbana-Champaign, Vandoeuvre-lès-Nancy, France; [6]Department of Physics, University of Illinois at Urbana-Champaign, Urbana, United States

**Abstract** The HIV-1 gp120/gp41 trimer undergoes a series of conformational changes in order to catalyze gp41-induced fusion of viral and cellular membranes. Here, we present the crystal structure of gp41 locked in a fusion intermediate state by an MPER-specific neutralizing antibody. The structure illustrates the conformational plasticity of the six membrane anchors arranged asymmetrically with the fusion peptides and the transmembrane regions pointing into different directions. Hinge regions located adjacent to the fusion peptide and the transmembrane region facilitate the conformational flexibility that allows high-affinity binding of broadly neutralizing anti-MPER antibodies. Molecular dynamics simulation of the MPER Ab-stabilized gp41 conformation reveals a possible transition pathway into the final post-fusion conformation with the central fusion peptides forming a hydrophobic core with flanking transmembrane regions. This suggests that MPER-specific broadly neutralizing antibodies can block final steps of refolding of the fusion peptide and the transmembrane region, which is required for completing membrane fusion.

***For correspondence:**
winfried.weissenhorn@ibs.fr

[†]These authors contributed equally to this work

**Competing interests:** The authors declare that no competing interests exist.

## Introduction

Viral fusion proteins catalyze virus entry by fusing the viral membrane with membranes of the host cell, thereby establishing infection. The HIV-1 envelope glycoprotein (Env) is a prototypic class I fusion protein that shares common pathways in membrane fusion with class II and III viral membrane fusion proteins (*Baquero et al., 2015*; *Harrison, 2008*; *Kielian and Rey, 2006*; *Weissenhorn et al., 2007*). HIV-1 Env is expressed as a gp160 precursor glycoprotein that is cleaved into the fusion protein subunit gp41 and the receptor-binding subunit gp120 by host furin-like proteases. Gp41 anchors Env to the membrane and associates non-covalently with gp120, thereby forming a stable trimer of heterodimers, the metastable Env prefusion conformation (*Chen, 2019*; *Wang et al., 2020*). Orchestration of a series of conformational changes transforms energy-rich prefusion Env into the low-energy, highly stable gp41 post-fusion conformation, which provides the free energy to overcome the kinetic barriers associated with bringing two opposing membranes into close enough contact to facilitate membrane fusion (*Harrison, 2008*; *Weissenhorn et al., 2007*).

HIV-1 gp41 is composed of several functional segments that have been shown or suggested to extensively refold upon fusion activation: the N-terminal fusion peptide (FP), a fusion peptide

proximal region (FPPR), the heptad repeat region 1 (HR1), a loop region followed by HR2, the membrane proximal external region (MPER), the transmembrane region (TMR), and a cytoplasmic domain. Structures of native Env trimers in complex with different broadly neutralizing antibodies revealed the conformation of the gp41 ectodomain lacking MPER in the native prefusion conformation (*Julien et al., 2013*; *Kwon et al., 2015*; *Lyumkis et al., 2013*; *Pancera et al., 2014*; *Stewart-Jones et al., 2016*; *Ward and Wilson, 2017*). Env interaction with CD4 results in opening of the closed prefusion trimer (*Liu et al., 2008*; *Ozorowski et al., 2017*), which includes the displacement of gp120 variable regions 1 and 2 (V1-V2) at the apex of the trimer and some changes in gp41 including extended and more compact HR1C helices, helical FP conformations (*Wang et al., 2018*) and accessibility of the MPER epitope for 10E8 interaction (*Lee et al., 2016*; *Rantalainen et al., 2020*). This is required for the formation of a stable ternary complex of Env-CD4 with the co-receptor (*Chang et al., 2005*; *Dobrowsky et al., 2008*; *Shaik et al., 2019*). Co-receptor-binding positions prefusion gp41 closer to the host-cell membrane (*Chen, 2019*) and induces a cascade of conformational changes in gp41. First, FP is repositioned by ~70 Å (*Pancera et al., 2014*) to interact with the target cell membrane, generating a 110 Å extended fusion-intermediate conformation (*Frey et al., 2008*; *Lai et al., 2014*) that bridges the viral and the host cell membrane (*Weissenhorn et al., 1999*). Subsequent refolding of HR2 onto HR1 leads to the formation of the six-helix bundle core structure (*Caffrey et al., 1998*; *Chan et al., 1997*; *Weissenhorn et al., 1997*), which pulls the viral membrane into close apposition to the host-cell membrane to set the stage for membrane fusion (*Weissenhorn et al., 1997*).

Membrane fusion generates a lipid intermediate hemifusion state, that is predicted to break and evolve to fusion pore opening (*Chernomordik and Kozlov, 2005*), which is regulated by six-helical bundle formation (*Markosyan et al., 2003*; *Melikyan et al., 2000*). Furthermore, residues within FPPR, FP, MPER, and TM have been as well implicated in fusion (*Bellamy-McIntyre et al., 2007*; *Long et al., 2011*; *Peisajovich et al., 2000*; *Salzwedel et al., 1999*; *Shang et al., 2008*) indicating that final steps in fusion are controlled by the conformational transitions of the membrane anchors into the final post-fusion conformation.

Here, we set out to understand the conformational transitions of the gp41 membrane anchors. We show that the presence of the membrane anchors increases thermostability. However, complex formation with a MPER-specific neutralizing nanobody stabilized an asymmetric conformation of the membrane anchors, which may represent a late fusion intermediate. We show that this conformation can be targeted by MPER bnAbs consistent with the possibility that MPER-specific nAbs can interfere all along the fusion process until a late stage. Starting from the asymmetric conformation, we used MD simulation based modelling to generate the final post-fusion conformation, which reveals a tight helical interaction of FP and TM in the membrane consistent with its high thermostability. In summary, our work elucidates the structural transitions of the membrane anchors that are essential for membrane fusion, which can be blocked by MPER-specific bnAbs up to a late stage in fusion.

## Results

### Gp41FP-TM interaction with 2H10

Two gp41 constructs, one containing residues 512 to 594 comprising FP, FPPR, and HR1 (N-terminal chain, chain N) and one coding for resides 629 to 716 including HR2, MPER and TM (C-terminal chain, chain C) (*Figure 1A*) were expressed separately, purified and assembled into the monodisperse trimeric complex gp41FP-TM (*Figure 1—figure supplement 1A*). Gp41FP-TM reveals a thermostability of ~93°C as measured by circular dichroism (*Figure 1—figure supplement 2A*) indicating that the presence of FP and TMR increases the thermostability by ~7°C compared to gp41 lacking FP and TM (*Buzon et al., 2010*). In order to facilitate crystallization, gp41FP-TM was complexed with the llama nanobody 2H10 (*Lutje Hulsik et al., 2013*) in β-OG buffer and purified by size exclusion chromatography (SEC) (*Figure 1—figure supplement 1B*). To determine the stoichiometry of binding, we performed isothermal titration calorimetry (ITC), which indicated that gp41FP-TM and 2H10 form a 3:1 complex with a $K_D$ of 2.1 ± 0.9 μM (*Figure 1—figure supplement 2B*). Interaction of gp41FP-TM with 2H10 was further confirmed by biolayer interferometry (BLI) analysis (*Figure 1—figure supplement 2C*).

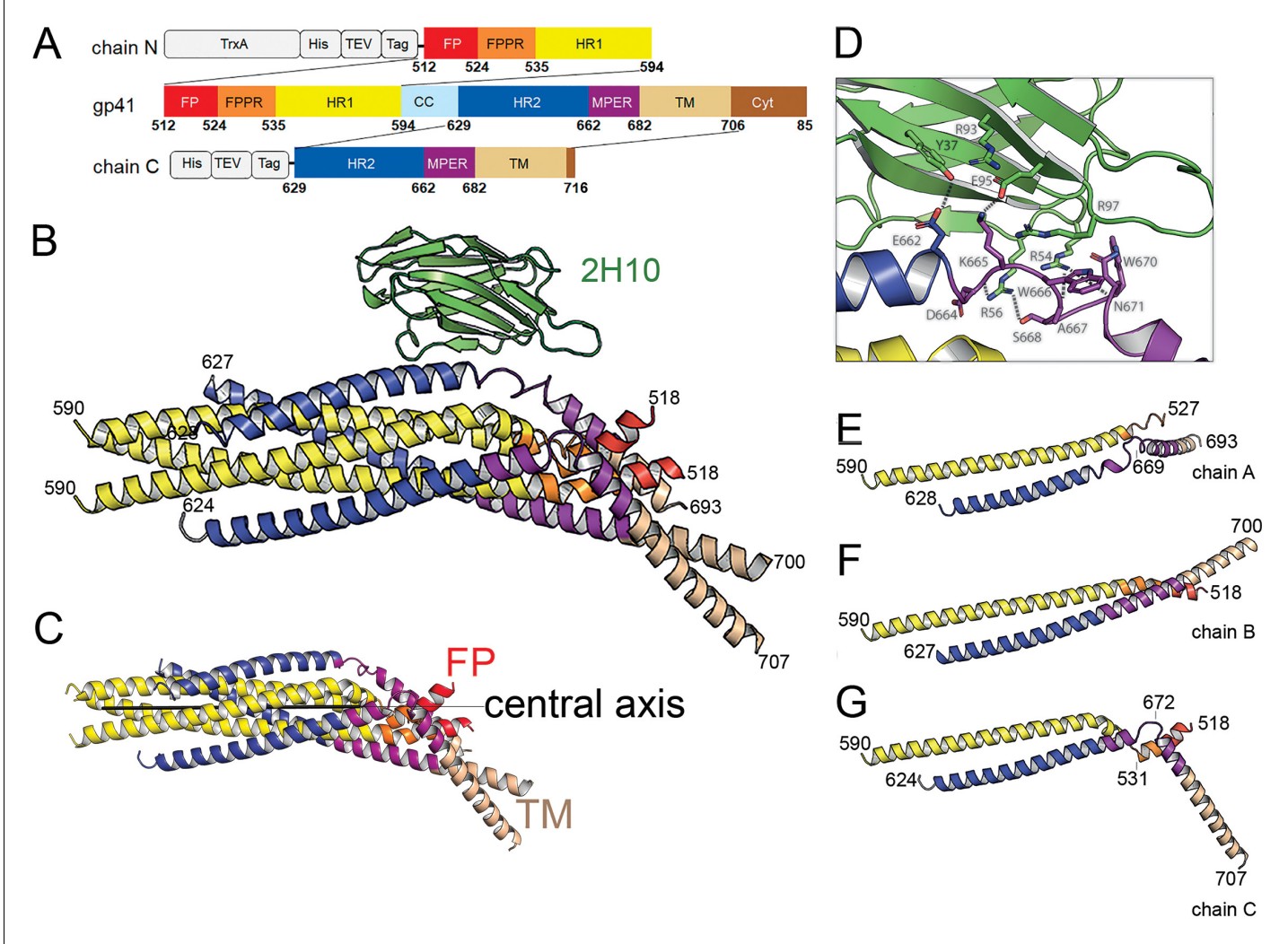

**Figure 1.** Crystal structure of gp41FP-TM in complex with 2H10. (A) Schematic drawing of gp41 and expression constructs of gp41 chains N and C. Sequence numbering is based on the HIV-1-HBX2 envelope gp160 sequence. Color coding is as follows: FP, fusion peptide, red; FPPR, fusion peptide proximal region, orange; HR1, heptad repeat region 1, yellow; HR2, heptad repeat region 2, blue; MPER, membrane proximal external region, violet; TM, transmembrane region, beige; CC, cys loop region, light blue and cyt, cytoplasmic domain brown. Expression tags used are TrxA, thioredoxin fusion protein, His, His-tag, TEV, TEV protease cleavage sequence, Tag, chain N contains a Flag-tag (DYKDDDDK sequence) and chain C an N-terminal enterokinase cleavage site (DDDDK). (B) Ribbon presentation of gp41TM-FP in complex with 2H10. Color-coding of the different segments is as indicated in the gp41 scheme (A), the 2H10 nanobody is colored in green. (C) Ribbon presentation of gp41TM-FP including the core six-helical bundle trimer axis (black line) revealing the different orientations of FP and TM. (D) Close-up of the interaction of gp41FP-TM with 2H10. Residues in close enough contact to make polar interactions are shown as sticks. (E, F, G) Ribbon diagram of the individual protomers named chain A, B, and C. Residues within the FPPR and MPER hinge regions are indicated.

The online version of this article includes the following figure supplement(s) for figure 1:

**Figure supplement 1.** Characterization of gp41 containing FP and TM.

**Figure supplement 2.** Biophysical characterization of gp41FP-TM and MPER Ab interaction.

**Figure supplement 3.** Close-ups of the model and its corresponding electron density.

**Figure supplement 4.** Comparison of the gp41FP-TM structure with gp41 core structures.

**Figure supplement 5.** Positioning of gp41FP-TM-2H10 in a bilayer by MD simulation.

**Figure supplement 6.** Crystal lattice packing.

**Figure supplement 7.** Membrane interaction of 2H10.

## Crystal structure of gp41 in complex with 2H10

The structure of gp41FP-TM in complex with 2H10 was solved by molecular replacement to a resolution of 3.8 Å (*Figure 1—figure supplement 3*; *Table 1*). The asymmetric unit contained trimeric gp41FP-TM bound to one 2H10 nanobody as indicated by ITC (*Figure 1—figure supplement 2B*). The six-helix bundle structure composed of three N-terminal and three C-terminal chains is conserved from HR1 residue A541 to HR2 residue L661 in all three protomers, and identical to previous structures (*Chan et al., 1997*; *Weissenhorn et al., 1997*). However, TMR and FP do not follow the three-fold symmetry and their chains point into different directions (*Figure 1B and C*). 2H10 interacts with chain C-A (*Figure 1B and D*) and stabilizes a partially extended MPER conformation, including a kink at L669 that positions the rest of MPER and TM (N674 to V693) at a 45° angle with respect to the six-helix threefold symmetry axis. The corresponding N-terminal chain A (chain N-A) has its FP disordered and FPPR from G527 to A533 is flexible, while the remaining FPPR and HR1 form a continuous helix (*Figure 1E*). The protomer A chain C 2H10 epitope spans from residues Q658 to N671, which is involved in a series of polar contacts with 2H10 covering a total buried surface of 712.2 Å$^2$. These include interactions of gp41FP-TM E662 to 2H10 Y37, S668 and the carbonyl of D664 to R56, K665 to E95, N671 and the carbonyl of A667 to R54, K655 to R97 and R93 contacts E95 to position it for interaction with K665 (*Figure 1D*). Notably, mutations of R56, R93, E95, and R97 have been shown to affect interaction with gp41 (*Lutje Hulsik et al., 2013*). Chain N-B of the second protomer forms a long continuous helix comprising FP, FPPR and HR1 from residues L518 to D589 with the first six residues of FP being disordered. Likewise, chain C-B folds into a continuous helix from M629 to A700 comprising HR2, MPER and TM (*Figure 1F*). Cα superimposition of chain C-B with MPER containing gp41 structures (*Buzon et al., 2010*; *Shi et al., 2010*) yields root mean-square deviations of 0.55 Å and 0.29 Å (*Figure 1—figure supplement 4*), indicating that the straight helical conformation is the preferred conformation in threefold symmetrical gp41. In the third protomer, chain N-C has a helical FP linked by flexible residues G531 to A533 to a short helix of FPPR that bends at A541 with respect to helical HR1. Its corresponding chain C-C contains helical HR2 and a flexible region from N671 to N674, which stabilizes a ~ 45° rotation of the remaining MPER-TM helix that extends to residue R707 (*Figure 1G* and *Figure 1—figure supplement 3B*). Thus, the structure reveals flexible regions within FPPR and MPER. FPPR flexibility is supported by strictly conserved G528 and G531, while MPER has no conserved glycine residues. However, the same kink within L661 to F673 has been observed in the MPER peptide structure (*Sun et al., 2008*), and in complex with bnAb 10E8 (*Huang et al., 2012*). The N-terminal FP residues 512 to 517 are disordered within the detergent micelle. Flexibility of this region in the absence of membrane is consistent with NMR peptide structures that propose a flexible coil structure of the N-terminal part of FP in solution followed by a helical region starting at L518 (*Serrano et al., 2017*) as observed here. Based on the flexible linkage of FP and TMR, we propose that both FPPR and MPER act as hinges during gp41 refolding leading to membrane fusion.

## MD simulation of gp41FP-TM in a lipid bilayer

In order to test whether the structure is influenced by the presence of the detergent, we probed its stability by MD simulation in a bilayer having the lipid composition of the HIV-1 envelope. This confirmed that the structure is stable in a membrane environment during a 1 μs simulation as only the flexibly linked FP of chain N-C moves within the bilayer during the simulation (*Figure 1—figure supplement 5A*). The tip of the 2H10 CDR3 dips into the bilayer (*Figure 1—figure supplement 5B*), hence confirming the membrane-anchoring role of W100 for neutralization (*Lutje Hulsik et al., 2013*).

## Neutralization activity of 2H10 depends on membrane interaction

The structure suggests that 2H10 stabilized asymmetry within the membrane anchors. Crystal packing effects on the conformation of the membrane anchors can be excluded since the orientation of the membrane anchors is likely not influenced by interprotomer contacts (*Figure 1—figure supplement 6*). In order to test whether stabilizing the asymmetric conformation is a feature of neutralizing MPER Abs, we further evaluated 2H10 as a neutralizing nanobody. Previously, 2H10 showed only modest neutralization as a bi-head (bi-2H10), whereas neutralization depended on W100 located at the tip of CDR3 (*Lutje Hulsik et al., 2013*), a hall mark of MPER-specific bnAbs (*Cerutti et al.,*

**Table 1.** Crystallographic data collection and refinement statistics.

| Data collection | Gp41FP-TM* |
| --- | --- |
| Space group | C 2 2 2$_1$ |
| Cell dimensions | |
| a, b, c (Å) | 96.75, 101.41, 234.42 |
| α, β, γ (°) | 90, 90, 90 |
| Resolution (Å) | 48.38–3.8 (3.94–3.8) * |
| Unique reflexions | 11179 (631)* |
| $R_{merge}$† | 0.23 (1.508)* |
| $R_{p.i.m}$ ‡ | 0.081 (0.548) |
| I / σI | 4.75 (1.74) * |
| Completeness (%) | 78.01 (54.69) * |
| Multiplicity | 9.1 (9.6) * |
| CC (1/2) | 0.992 (0.628) * |
| Refinement | |
| Resolution (Å) | 48.38–3.8 (3.936–3.8)* |
| No. reflections | 9154 (630)* |
| Reflections used for $R_{free}$§ | 550 (51)* |
| $R_{work}$§ / $R_{free}$** | 0.265/0.308 |
| No. atoms | |
| Protein | 4440 |
| Ligand/ion | 0 |
| Water | 0 |
| Wilson B (Å$^2$) | 75.8 |
| Average B-factors (Å$^2$) | |
| Overall | 91.76 |
| Protein | 91.76 |
| Ligand/ion | |
| Water | |
| R.m.s deviations | |
| Bond lengths (Å) | 0.003 |
| Bond angles (°) | 0.66 |
| Ramachandran Plot (%) | |
| Favored | 96.65 |
| Outliers | 0.37 |
| PDB ID | 7AEJ |

*Data collected from two crystals were used for structure determination.

The statistics are for data that were truncated by STARANISO to remove poorly measured reflections affected by anisotropy. $R_{merge}$, $R_{p.i.m}$ and multiplicity are calculated on unmerged data prior to STARANISO truncation. For comparison, after STARANISO truncation, $R_{merge}$ in the resolution shell 3.97 Å - 3.85 Å is 0.787.

† Parentheses refer to outer shell statistics.

‡ $R_{merge} = \Sigma_{hkl} \Sigma_i \mid I_{hkl,i}- < I_{hkl} > \mid / \Sigma_{hkl} \Sigma_i I_{hkl,i}$, where $I_{hkl,i}$ is the scaled intensity of the ith measurement of reflection h, k, l, and $<Ihkl >$ is the average intensity for that reflection.

§ $R_{p.i.m.} = \Sigma_{hkl} \sqrt{1/(n-1)} \Sigma_i \mid I_{hkl,i}- < I_{hkl} > \mid / \Sigma_{hkl} \Sigma_i I_{hkl,i}$.

¶ $R_{work} = \Sigma_{hkl} \mid Fo - Fc \mid / \Sigma_{hkl} \mid Fo \mid \times 100$, where Fo and Fc are the observed and calculated structures factors.

** $R_{free}$ was calculated as for $R_{work}$, but on a test set of 5% of the data excluded from refinement.

The online version of this article includes the following source data for  Table 1:

**Source data 1.** Env pseudoviruses.

*2017*). In order to engineer breadth and potency of monovalent 2H10, we increased its potential membrane interaction capacity by changing CDR3 S100d to F (2H10-F) alone and in combination with additional basic residues S27R, S30K, and S74R (2H10-RKRF) within the putative 2H10 membrane-binding interface suggested by MD simulation (*Figure 1—figure supplement 5C*). Wild type 2H10 did not show significant neutralization against a panel of 10 clade B pseudo-viruses as reported previously (*Lutje Hulsik et al., 2013*), with the exception of some weak neutralization of NL4-3 and SF163P3. However, both 2H10-F and 2H10-RKRF show improved potency and breadth neutralizing six and eight pseudo-viruses, respectively, albeit with less potency than wild-type bi-2H10 and bnAb 2F5, the latter recognizing an overlapping epitope (*Table 2*). Notably, enhancing the membrane interaction surface did not increase the detection of non-specific membrane binding in vitro (*Figure 1—figure supplement 7*). This result confirms monovalent 2H10 as a modest anti-MPER Ab that neutralizes by engaging MPER and the membrane.

## 2H10 blocks fusion before the stage of lipid mixing

The efficacy of bi-2H10 and 2H10-RKRF for blocking membrane merging was further assessed in peptide-induced lipid-mixing assays (*Figure 2*). In the experimental setting (*Apellániz et al., 2014b*), a vesicle population is primed for fusion by addition of the N-MPER peptide containing the 2H10 epitope, which produces a fluorescence intensity spark at time 20 s (*Figure 2A*). Under these experimental conditions, incorporation of the peptide into the vesicles takes less than 10 s. After 120 s, the mixture is supplemented with target vesicles fluorescently labeled with N-NBD-PE/N-Rh-PE (indicated by the arrow in *Figure 2A*). The increase in NBD intensity as a function of time follows the mixing of the target vesicle lipids with those of the unlabeled vesicles (kinetic trace labeled '+N-MPER'), a process not observed when labeled target vesicles are injected in a cuvette containing unlabeled vesicles not primed with peptide ('no peptide' trace).

MPER antibodies have been shown to interact directly with membrane-bound peptide epitopes, thereby changing their insertion state (depth and oligomerization levels) and inhibiting their capacity to induce fusion (*Lorizate et al., 2006*; *Sun et al., 2008*). Accordingly, lipid mixing was strongly attenuated when the vesicles primed for fusion with N-MPER were incubated with bi-2H10 before addition of the target vesicles (*Figure 2A*,+N-MPER/+bi-2H10, dotted trace). Thus, the N-MPER-induced membrane perturbations, which can induce fusion with target membranes, were inhibited by incubation with bi-2H10. Comparison of the kinetics of the lipid-mixing blocking effect of 2H10-RKRF, bi-2H10 and Fabs 2F5 and 10E8 showed that the four antibodies inhibited both the initial rates and final extents of lipid mixing induced by N-MPER (*Figure 2B*). Using a control MPER peptide lacking the 2H10 and 2F5 epitopes for vesicle priming no inhibition of lipid mixing by 2H10-RKRF, bi-2H10 and 2F5 Fab was observed while 10E8 still blocked lipid mixing, (*Figure 2C*) corroborating that the inhibitory effects depend on epitope recognition. The use of the 10E8 Fab as an additional control further supports the blocking effect mediated by epitope recognition. Notably, Fab 10E8 inhibits lipid mixing in a concentration-dependent manner (*Figure 2—figure supplement 1A*).

**Table 2.** Pseudovirus neutralization by 2H10, 2H10-F, 2H10-RKRF, and bi-2H10 in comparison to 2F5 and VRC01. IC50s are indicated in µg/ml.

|  | Tier | 2H10 wt | 2H10-F | 2H10-RKRF | Bi-2H10 | 2F5 | VRC01 |
|---|---|---|---|---|---|---|---|
| NL4-3 | 1 | 25.20 | 18.68 | 9.15 | 1.84 | 0.16 | 0.20 |
| MN-3 | 1 | >50.00 | 30.38 | 9.36 | 1.39 | 0.03 | 0.06 |
| BaL.26 | 1 | >50.00 | 19.38 | 9.63 | 6.05 | 1.21 | 0.13 |
| SF162 | 1a | >50.00 | >50.00 | 25.19 | 6.14 | 1.22 | 0.39 |
| SF162P3 | 2 | 22.04 | 13.14 | 6.76 | 1.32 | 1.96 | 0.24 |
| SC422661.8 | 2 | >50.00 | >50.00 | 27.93 | 3.79 | 1.00 | 0.27 |
| JR-FL | 2 | 44.65 | 16.93 | 6.95 | 1.49 | 0.97 | 0.11 |
| JR-CSF | 2 | >50.00 | 21.66 | 10.85 | 2.85 | 1.24 | 0.37 |
| QH0692.42 | 2 | >50.00 | >50.00 | >50.00 | >50.00 | 1.20 | 1.21 |
| THRO4156.18 | 2 | >50.00 | >50.00 | >50.00 | >50.00 | >50.00 | 3.84 |

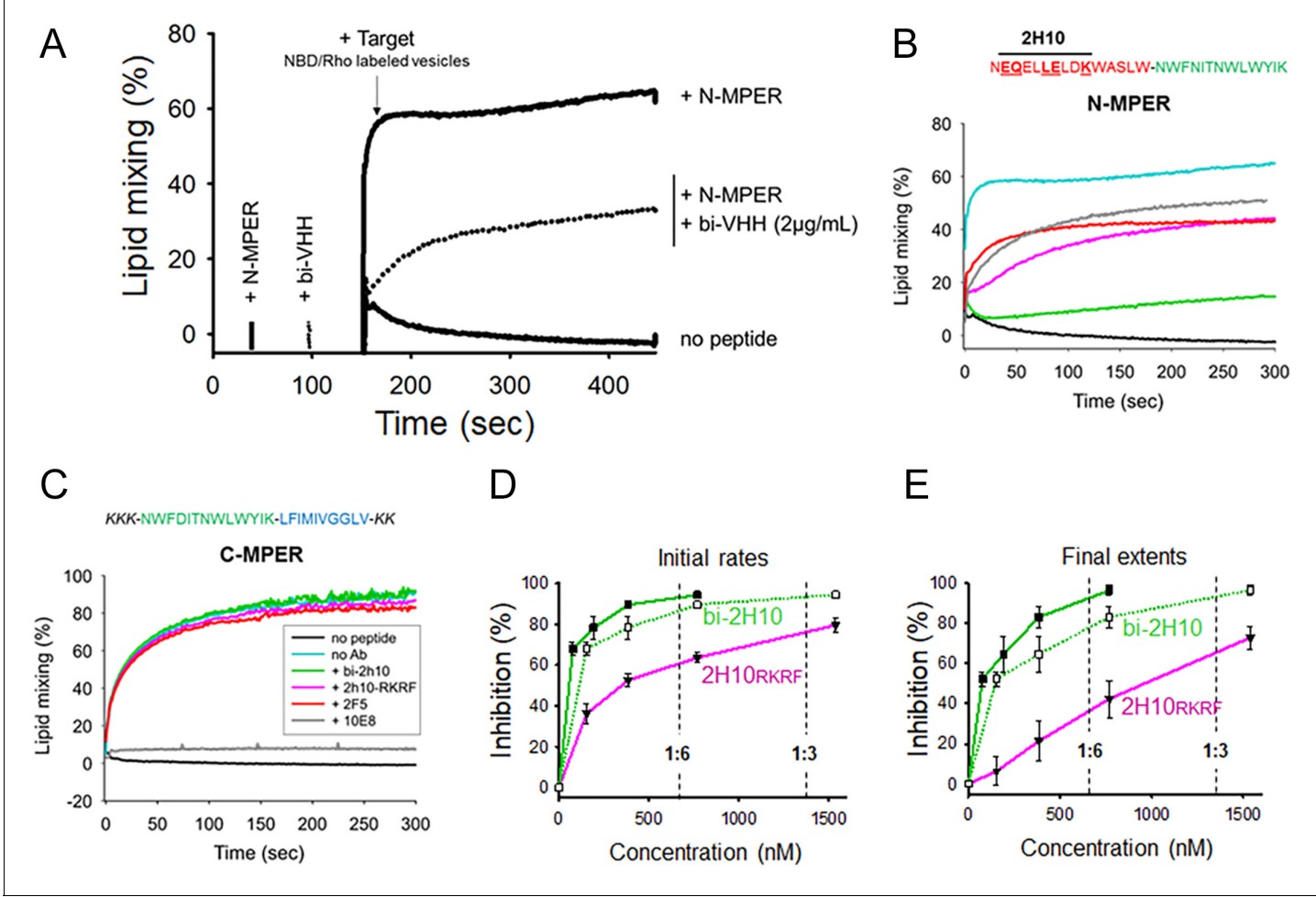

**Figure 2.** Vesicle-vesicle fusion inhibition by 2H10-RKRF, bi-2H10, 2F5, and 10E8. (**A**) Time course of the lipid-mixing assay using fusion-committed vesicles. At time 30 s ('+N-MPER'), peptide (4 μM) was added to a stirring solution of unlabeled vesicles (90 μM lipid), and, after 120 s (indicated by the arrow), the mixture was supplemented with N-NBD-PE/N-Rh-PE-labeled vesicles (10 μM lipid). The increase in NBD fluorescence over time follows the dilution of the probes upon mixing of lipids of target and primed vesicles (+N MPER trace). NBD increase was substantially diminished in samples incubated with bi-2H10 (2 μg/ml) prior to the addition of the target vesicles (+bi-2H10, dotted trace), and totally absent if unlabeled vesicles were devoid of peptide ('no peptide' trace). (**B**) Kinetic traces of N-MPER-induced lipid-mixing comparing the blocking effects of 2H10-RKRF, bi-2H10, Fab 2F5 and Fab 10E8. (**C**) Absence of effects on lipid-mixing of 2H10-RKRF, bi-2H10 and Fab 2F5 when vesicles were primed for fusion with the C-MPER peptide, devoid of 2H10 and 2F5 epitope sequences. The positive control Fab 10E8 efficiently blocked the process. Antibody concentrations were 20 μg/ml in these assays. (**D**) Dose-response plots comparing the inhibitory capacities of 2H10-RKRF and bi-2H10 (purple and green traces, respectively). Levels of lipid-mixing 20 or 300 s after target vesicle injection were measured (initial rates **D** and final extents, **E**) and percentages of inhibition calculated as a function of the Ab concentration. The dotted line and empty symbols correspond to the effect of bi-2H10 when the concentration of the component 2H10 was plotted. The slashed vertical lines mark the 2H10-to-peptide ratios of 1:6 and 1:3. Plotted values are means ± SD of three independent experiments.

The online version of this article includes the following figure supplement(s) for figure 2:

**Figure supplement 1.** Vesicle-vesicle fusion inhibition by 10E8 Fabs.

Furthermore, inhibition of lipid mixing depends on direct membrane interaction since increasing hydrophobicity within the 10E8 Fab-membrane interface increases the inhibitory effect and decreasing 10E8 hCDR3 hydrophobicity abrogates lipid mixing (*Figure 2—figure supplement 1B and C*), which is in agreement with neutralization determinants of 10E8 (*Chen et al., 2014*; *Rujas et al., 2018*). The complete 10E8 epitope encompasses C-terminal MPER residues plus N-terminal TM residues, the latter not included in the sequence covered by the N-MPER peptide. Consistent with this recognition pattern, 10E8 blocked C-MPER-induced fusion more efficiently than N-MPER-induced fusion, but was effective against both peptides. The capacity for blocking the lipid-mixing process

also correlates with the neutralization potency of MPER bnAbs (*Apellániz et al., 2014b*). Following this trend, fusion inhibition levels estimated as a function of the antibody concentration confirmed the apparent higher potency exhibited by the bi-2H10 (*Figure 2D*). Lower concentrations of bi-2H10 compared to 2H10-RKRF were required to attain full blocking of the lipid-mixing process when measured 20 s (initial rates) or 300 s (final extents) after target-vesicle injection (*Figure 2D and E*). The higher inhibitory potency of bi-2H10 indicates an avidity effect, which was also evident when the concentration of the epitope-binding fragments was plotted (*Figure 2D and E*, empty squares and dotted line). Moreover, bi-2H10 appeared to block the process efficiently even at 2H10:N-MPER ratios below 1:3 (mol:mol), consistent with the involvement of peptide oligomers in the promotion of membrane fusion. Based on these data, we suggest that both 2H10-RKRF and bi-2H10 neutralize HIV-1 at the stage of lipid mixing.

## GP41FP-TM interaction with MPER bnAbs

Although the 2H10 epitope overlaps with the 2F5 MPER epitope (*Ofek et al., 2004*), the 2F5-bound peptide structure (*Ofek et al., 2004*), cannot be superimposed without major clashes with adjacent gp41 protomers. In contrast, Cα superposition of the structures of 10E8 and LN01 in complex with MPER peptides demonstrated possible binding to gp41FP-TM chain C-C (*Figure 3A and B*). Furthermore, HCDR3 of both 10E8 and LN01 could make additional hydrophobic contacts with adjacent FP in this binding mode. To confirm 10E8 and LN01 interaction, we performed immunoprecipitation of gp41FP-TM with both bnAbs, which confirmed their interaction in vitro (*Figure 3—figure*

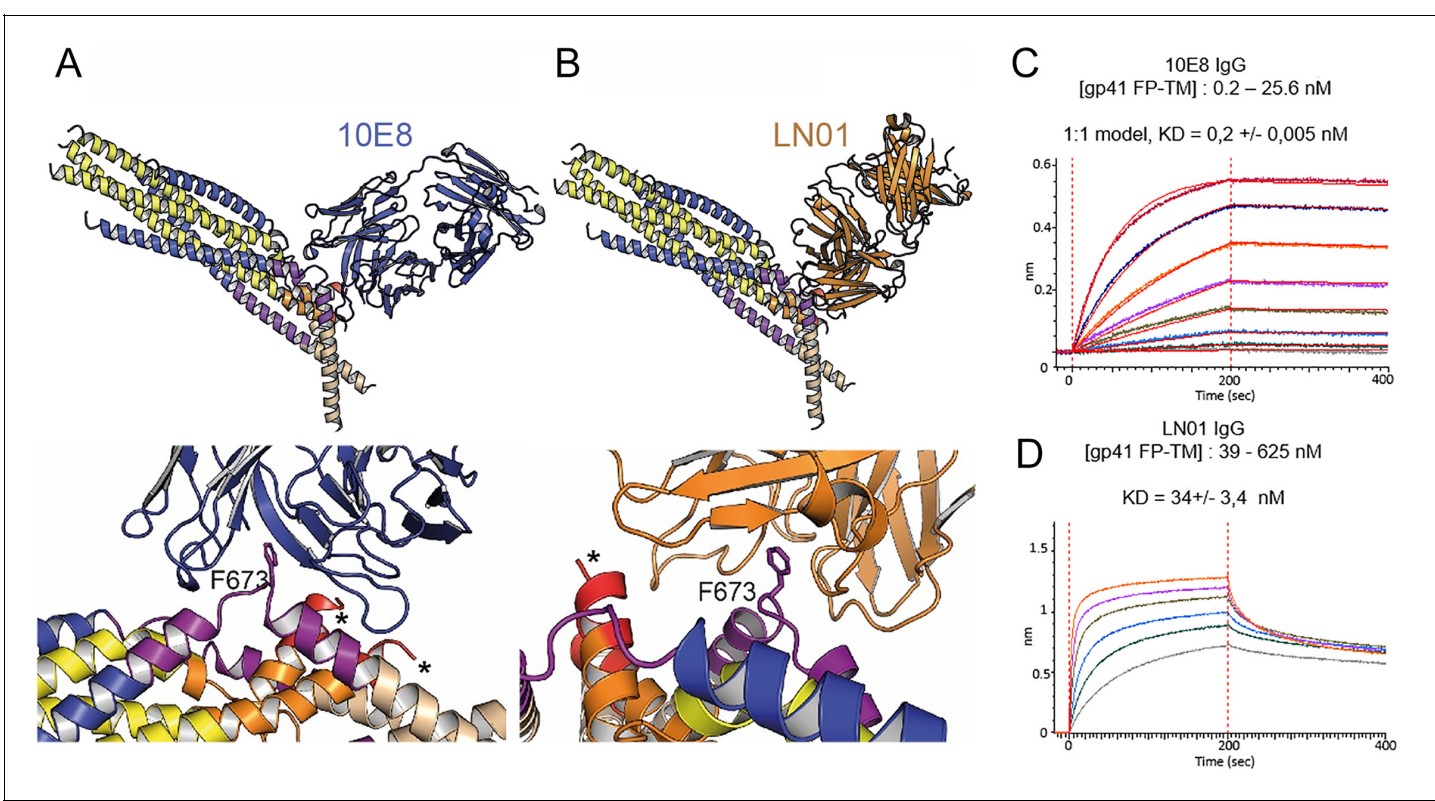

**Figure 3.** Gp41FP-TM interaction with bnAbs LN01 and 10E8. (**A**) Cα superposition of the MPER peptide structure in complex with LN01 (pdb 6snd) onto chain C-C of the gp41FP-TM-2H10 structure. The lower panel shows a close-up of the interaction oriented with respect to gp41 F673. (**B**) Cα superposition of the MPER peptide structure in complex with 10E8 (pdb 5iq7) onto the corresponding chain C-C of gp41FP-TM. The lower panel shows a close-up of the interaction in the same orientation as in A. (**C**) Bio-layer interferometry (BLI) binding of gp41FP-TM to 10E8 and (**D**) to LN01. 10E8 binding was fit to 1:1 model and for LN01 a steady state model was employed for fitting the data. For 10E8 binding, gp41FP-TM was used at concentrations from 0.2 to 25.6 nM and for LN01 binding gp41FP-TM concentrations ranged from 39 to 625 nM.

The online version of this article includes the following figure supplement(s) for figure 3:

**Figure supplement 1.** Pull down of gp41FP-TM by bnAb 10E8 and LN01.

*supplement 1*). We next validated binding by bio-layer interferometry (BLI) using gp41FP-TM as analyte. This revealed $K_D$s of 0,2 nM for 10E8 and 34 nM for LN01 (*Figure 3C and D*). We conclude that bnAbs 10E8 and LN01 interact with gp41FP-TM with high affinity which may also stabilize an asymmetric gp41 conformation similar to the one observed in complex with 2H10 as suggested by the structural modeling (*Figure 3A and B*).

## Building a post-fusion conformation by MD simulation

In order to follow the final refolding of the membrane anchors, we modeled the post fusion conformation employing MD simulation. Assuming that the final post-fusion conformation shows a straight symmetric rod-like structure we constructed a model of gp41 from the protomer composed of the straight helical chains N-B and C-B (*Figure 4—figure supplement 1A and B*). This conformation is also present in the symmetric six-helix bundle structures containing either MPER (*Shi et al., 2010*) or FPPR and MPER (*Buzon et al., 2010*; *Figure 1—figure supplement 3*). In this model, FP and TM do not interact tightly (*Figure 4—figure supplement 1B*), which, however, does not explain the increased thermostability in the presence of FP and TM (*Figure 1—figure supplement 2A*). One-µs MD simulation of this model (*Figure 4—figure supplement 1B*) in solution, rearranges the membrane anchors such that they adopt a compact structure with trimeric FP interacting with adjacent TMs. Furthermore, the TMs kink at the conserved Gly positions 690 and 691, as observed previously (*Pinto et al., 2019*; *Figure 4—figure supplement 1C*). In order to recapitulate the stability of the model in the membrane, we performed an additional 1-µs MD simulation of the model (*Figure 4—figure supplement 1C*) in a bilayer resembling the HIV-1 lipid composition, which relaxed the TM to its straight conformation (*Figure 4A*). The final structural model reveals tight packing of trimeric FP flexibly linked to HR1 by FPPR G525 to G527 (*Figure 4B*). HR2-MPER and TMR form continuous helices with the TMRs packing against trimeric FP (*Figure 4A and C*), which spans one monolayer (*Figure 4A*). As conserved tryptophan residues within MPER have been previously implicated in fusion (*Bellamy-McIntyre et al., 2007*; *Salzwedel et al., 1999*), we analyzed their structural role in the post fusion model. This reveals that the indole ring of W666 is sandwiched between Leu669 and T536 and packs against L537. W670 makes a coiled-coil interaction with S534, while W672 is partially exposed and packs against L669 and T676. W678 binds into a hydrophobic pocket defined by I675, L679, I682 and adjacent FP/FPPR residues F522 and A526. W680 is partially exposed, but reaches into a pocket created by the flexible FPPR coil (*Figure 4—figure supplement 2*). We therefore propose that most of the tryptophan residues have structural roles in the post-fusion conformation, hence providing an explanation for their functional role in fusion (*Salzwedel et al., 1999*). The MPER epitopes recognized by 10E8 and LN01 are exposed in the post-fusion model, but antibody docking to this conformation produced major clashes, consistent with no expected binding to the final post fusion conformation.

## Structural transitions of gp41

A number of Env SOSIP structures revealed the native conformation of gp41 (*Figure 5A, B and C*; *Kumar et al., 2019*; *Pancera et al., 2014*; *Yuan et al., 2019*). The gp41FP-TM crystal structure and the model of its post-fusion conformation provide further insight into the path of conformational changes that native gp41 must undergo to adopt its final lowest energy state conformation. The first major conformational changes in gp41 that take place upon receptor binding are extension of HR1 and FPPR into a triple stranded coiled coil with flexible linkers connecting FPPR to FP that may initially interact laterally with the external leaflet of the bilayer thereby projecting FP ~115 Å away from its starting position (*Figure 5D*). Notably, such an early intermediate fusion conformation structure has been reported for influenza hemagglutinin (HA) (*Benton et al., 2020*). This is likely followed by an extension and rearrangement of HR2 and MPER producing 11–15 nm long intermediates that connect the viral and cellular membranes (*Ladinsky et al., 2020*; *Lai et al., 2014*). Gp41 refolding into the six-helix bundle structure then produces flexibly linked asymmetric conformations of FPPR-FP and MPER-TM anchored in the cellular and viral membranes, respectively, as indicated by the gp41FP-TM structure. This intermediate conformation may bring viral and cellular membranes into close proximity (*Figure 5E*) or may act at the subsequent stage of hemifusion (*Figure 5F*). Further refolding and interaction of FP-FPPR and MPER-TM may generate the stable post fusion conformation (*Figure 5G*), a process that completes membrane fusion. Notably, in this model based on the

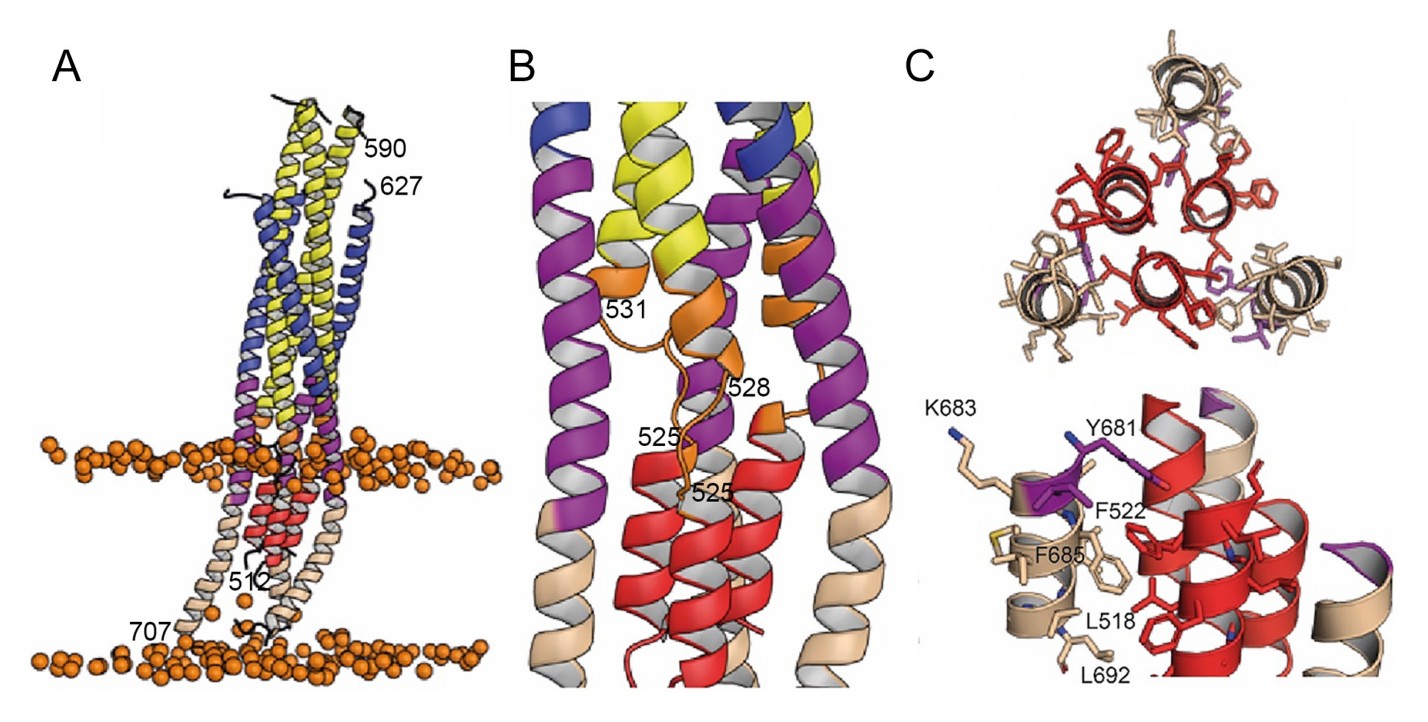

**Figure 4.** Interactions within the final post fusion conformation of gp41FP-TM modeled by MD. (**A**) Model of gp41FP-TM (*Figure 1—figure supplement 7C*) after 1μs MD simulation in a bilayer. Phosphate groups of the phospholipids are shown as orange spheres to delineate the membrane boundaries. (**B**) Close-up on the MPER and FPPR flexible regions. (**C**) Close-up of the interaction of FP (residues 514–524) and TM (residues 681–692) viewed along the three-fold axis from the N-terminus indicating an intricate network of hydrophobic interactions (left panel) and from the side (right panel). Interacting side chains are labeled and shown as sticks.

The online version of this article includes the following figure supplement(s) for figure 4:

**Figure supplement 1.** Modeling a post fusion conformation by MD simulation.

**Figure supplement 2.** Positions of the conserved tryptophan residues of MPER in the post fusion model.

crystal structure, FP spans only one leaflet of the lipid bilayer facilitating local lipid head group interactions with the N-terminus of FP.

## Discussion

Membrane fusion is an essential step of infection for enveloped viruses such as HIV-1, and requires extensive conformational rearrangements of the Env prefusion conformation (*Julien et al., 2013*; *Lyumkis et al., 2013*; *Pancera et al., 2014*) into the final inactive post-fusion conformation (*Harrison, 2008*; *Weissenhorn et al., 2007*). The fusion model predicts that six-helix bundle formation apposes viral and cellular membranes with FP and TM inserted asymmetrically in the cellular membrane and the viral membrane, respectively (*Weissenhorn et al., 1999*). Here, we show that gp41 containing its membrane anchors can adopt this predicted conformation, which is facilitated by flexible hinges present in FPPR and MPER corroborating their essential roles in membrane fusion (*Blumenthal et al., 2012*; *Chen, 2019*; *Harrison, 2008*). The asymmetric conformation of the membrane anchors suggest further that bundle formation occurs before pore formation as suggested previously (*Markosyan et al., 2003*; *Melikyan et al., 2000*). The membrane-fusion model proposes further that final steps in fusion involves rearrangement and interaction of TM and FP (*Weissenhorn et al., 1999*), which is confirmed by the MD-simulation model of the post-fusion conformation. Furthermore, the length of the rod-like post-fusion structure of 13 nm lacking the C-C loop is consistent with the gp41 structure lacking FP and TM (*Weissenhorn et al., 1996*).

FP is helical in the gp41FP-TM-2H10 complex and the MD-based post-fusion conformation, in agreement with NMR-based helical FP peptide models (*Li and Tamm, 2007*; *Serrano et al., 2014*),

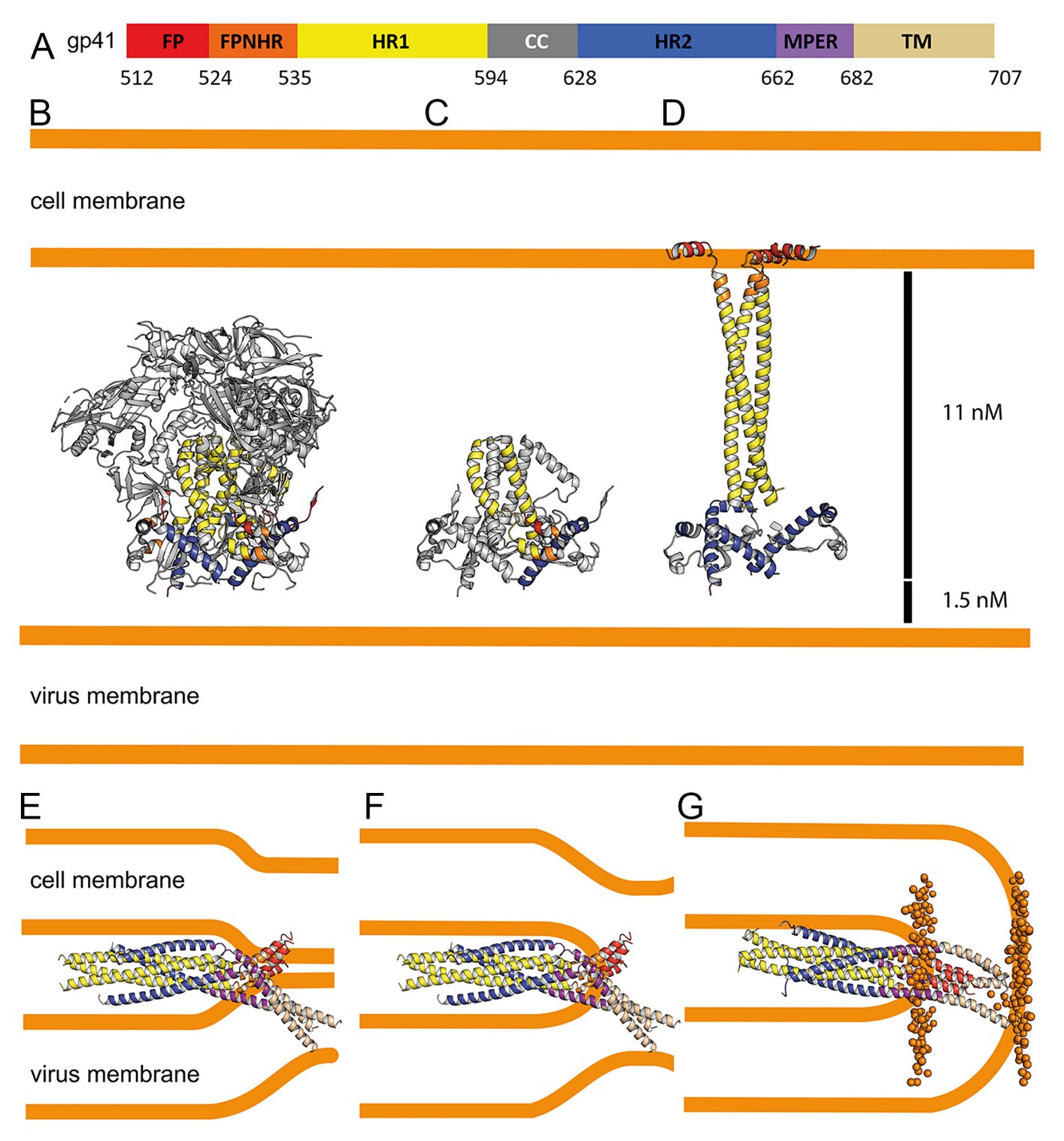

**Figure 5.** Conformational transitions of gp41 that lead to membrane apposition and membrane fusion. (**A**) Representation of the different domains of gp41 with the residue numbers delimiting each domain as indicated. The same color code has been used in all the figures. (**B**) Ribbon presentation of the Env prefusion conformation (pdb 5fuu), gp41 is constrained by gp120 in its native conformation. The structure of native gp41 lacks the MPER and TM regions. MPER is spanning a distance of 1.5 nm (*Li et al., 2020*). (**C**) Ribbon of native gp41, one chain is colored according to the sheme in A and the other two chains are shown in gray. (**D**) Binding to cellular receptors CD4 and subsequently to CXCR4/CCR5 induces a series of conformational changes that eventually leads to the dissociation of gp120. During this process, HR1, FPPR and FP will form a long triple stranded coiled coil extending 11 nm toward the target cell membrane. In a first step, HR2 may keep its prefusion conformation in analogy to a similar intermediate, activated

*Figure 5 continued on next page*

*Figure 5 continued*

influenza virus HA structure (**Benton et al., 2020**). Alternatively, HR2 may dissociate and form a more extended conformation in agreement with locked gp41 structures bridging viral and cellular membranes that bridge distances of 11–15 nm (**Ladinsky et al., 2020**). (E) Bending of HR1 and HR2 will result in the six-helical bundle core structure bringing cellular and viral membranes into close apposition with the three FPs anchored in the cellular membrane and the three TMs anchored in the viral membrane, the gp41 conformation represented by the gp41FP-TM structure. This intermedaite gp41 conformation may have brought both membranes into close apposition or may have already induced hemifuison as indicated in **F**. (G) Further reolding of FPPR-FP and MPER-TM results in the final extremely stable post-fusion conformation. This suggests that rearrangment of the membrane anchors plays crucial roles in lipid mixing, breaking the hemifusion diaphragm to allow fusion pore opening. Boundaries of the lipid layers are shown with orange spheres representing the phosphate atoms of the lipids present in the MD simulation (snapshot taken after 1μs MD simuation).

although β-strand structures of FP have been implicated in fusion as well (**Apellániz et al., 2014a**). In comparison, in native Env conformations, FP adopts multiple dynamic conformations that are recognized by broadly neutralizing antibodies (**Blattner et al., 2014**; **Kong et al., 2016**; **Kumar et al., 2019**; **Yuan et al., 2019**). In the post-fusion conformation, FP spans one monolayer of the membrane, in contrast to suggested amphipathic helix-like interaction of FP with the outer layer of the target cell membrane, which may constitute a first contact with the cellular membrane (**Han et al., 2001**; **Lai et al., 2012**).

Furthermore, the coiled-coil interactions within FP and with TM in the post-fusion model explain the increased thermostability of gp41FP-TM compared to gp41 lacking FP and TM (**Buzon et al., 2010**). We propose that refolding of FP and TM can liberate additional free energy to catalyze final steps of fusion. Hence, replacement of TM by a phosphatidylinositol (PI) anchor inhibits membrane fusion (**Salzwedel et al., 1993**; **Weiss and White, 1993**), akin to the GPI-anchored HA inhibition of influenza virus membrane fusion at the stage of hemifusion (**Kemble et al., 1994**).

Combinatorial mutagenesis of the conserved tryptophan residues within MPER was shown to affect syncytium formation, although individual mutations had no effect (**Muñoz-Barroso et al., 1999**; **Salzwedel et al., 1999**). However, in another study a W666A mutation was shown to affect infection (**Bellamy-McIntyre et al., 2007**), which was further enhanced in combination with a I675A mutation albeit without effect on syncytium formation (**Narasimhulu et al., 2018**). Although it is not known at which stage of the fusion process the concerted action of the tryptophan residues is required, our structural model suggests that W666, W670, and W678 likely play a structural role in stabilizing the post-fusion conformation. Furthermore, combinatorial mutations to leucine of the TM region comprising residues 683–693 affect virus infectivity (**Shang et al., 2008**) in agreement with the proposed interaction of this part of the TM with FP in the post fusion model. In addition, mutation of the conserved arginine at position 693 reduces fusion efficiency (**Long et al., 2011**), which however, may be related to its role in stabilizing a tilted TM conformation in the bilayer during the fusion process (**Chiliveri et al., 2018**; **Pinto et al., 2019**). Our structural model of the post-fusion conformation is further in agreement with proposed interactions of FPPR and MPER, as well as FP and TM (**Louis et al., 2016**; **Reuven et al., 2012**), thereby corroborating their essential roles at late stages of membrane fusion.

Gp41FP-TM interaction with the 2H10 MPER-specific nanobody stabilizes the asymmetric conformation of the membrane anchors. In order to confirm that 2H10 is, indeed, a neutralizing MPER-specific nanobody, we engineered increased 2H10 membrane binding, which improved breadth and potency of 2H10 neutralization, in agreement with enlarged potency by increasing membrane-binding of 10E8 (**Chen et al., 2014**; **Kwon et al., 2018**; **Rujas et al., 2018**). This confirmed 2H10 as a modest anti-MPER neutralizing antibody, whose neutralization potency is not only enhanced by using bi-2H10 with increased avidity (**Lutje Hulsik et al., 2013**), but also by increasing potential membrane interaction of the 2H10 monohead. Consistent with its neutralization capacity, 2H10 inhibits membrane fusion at the stage of lipid mixing like 2F5 and other anti-MPER bnAbs (**Apellániz et al., 2014b**; **Huarte et al., 2008**). Moreover, gp41FP-TM interacts with MPER bnAbs 10E8 (**Huang et al., 2012**) and LN01 (**Pinto et al., 2019**) in agreement with docking both structures onto the kinked protomer C chain C MPER epitope. Notably, the kink in the MPER peptide in complex with 10E8 (**Huang et al., 2012**) is similar to the protomer C chain C kink and present in MPER peptide NMR structures (**Serrano et al., 2014**; **Sun et al., 2008**). Furthermore, Ala mutations in the kink (671-674) affect cell-cell fusion and lower virus infectivity (**Sun et al., 2014**) corroborating the physiological relevance of the kinked conformation. We therefore propose that 10E8 and LN01

binding to gp41FP-TM stabilizes similar asymmetry, as observed in the gp41FP-TM-2H10 structure by sampling the dynamic states of the membrane anchors.

Our data indicate that MPER antibodies can act all along the gp41 refolding pathway from blocking initial conformations of close to native Env (*Carravilla et al., 2019*; *Lee et al., 2016*; *Rantalainen et al., 2020*) up to a late fusion intermediate state that has already pulled viral and cellular membranes into close apposition. This opens a long temporal window of action for MPER bnAbs consistent with the findings that the half-life of neutralization of MPER bnAbs is up to 30 min post virus exposure to target cells (*Shen et al., 2010*; *Williams et al., 2017*). Furthermore, only one Ab per trimer may suffice to block final refolding of the membrane anchors required for fusion. Finally, the presence of dynamic linkers connecting the core of viral fusion proteins with their membrane anchors FP and TM may be a general feature of all viral membrane fusion proteins including the proposed flexible linkage of influenza virus HA FP to the central HA2 coiled coil (*Benton et al., 2020*).

# Materials and methods

## Key resources table

| Reagent type (species) or resource | Designation | Source or reference | Identifiers | Additional information |
|---|---|---|---|---|
| Recombinant DNA reagent | Chain N pETM20 (plasmid) | This study | pETM20, PEPcore facility-EMBL | |
| Recombinant DNA reagent | Chain C pETM11 (plasmid) | This study | pETM11, PEPcore facility-EMBL | |
| Recombinant DNA reagent | 2H10 pAX51 (plasmid) | *Lutje Hulsik et al., 2013* | PMID:23505368 | |
| Recombinant DNA reagent | 2H10-F pAX51 (plasmid) | This study | | S100d to F mutation |
| Recombinant DNA reagent | 2H10-RKRF pAX51 (plasmid) | This study | | S27R, S30K, S74R and S100d to F mutations |
| Strain, strain background (*Escherichia coli*) | BL21(DE3) | ThermoFisher | Cat# EC0114 | |
| Strain, strain background (*Escherichia coli*) | C41(DE3) | Lucigen | Cat#60442 | |
| Cell line (*Homo sapiens*) | TZM-bl | NIH AIDS Reagent Program | Cat#ARP-8129 RRID:CVCL_B478 | |
| Cell line (*Homo sapiens*) | HEK293T | ATTC | Cat# CRL-11268 RRID:CVCL_1926 | |
| Antibody (human monoclonal) | α-gp41, LN01 | *Pinto et al., 2019* | PMID:31653484 | See Materials and methods |
| Antibody (human monoclonal) | α-gp41, 10E8 | *Huang et al., 2012* | PMID:23151583 RRID:AB_2491067 | See Materials and methods |
| Antibody (human monoclonal) | α-gp41, 2F5 | *Muster et al., 1993* | PMID:7692082 | See Materials and methods |
| Antibody (human monoclonal) | α-gp140, VRC01 | *Zhou et al., 2010* | PMID:20616231 RRID:AB_2491019 | See Materials and methods |

*Continued on next page*

Continued

| Reagent type (species) or resource | Designation | Source or reference | Identifiers | Additional information |
|---|---|---|---|---|
| Antibody (llama nanobody) | α-gp41, 2H10 | *Lutje Hulsik et al., 2013* | PMID:23505368 | See Materials and methods |
| Antibody (llama nanobody) | α-gp41, bi-2H10 | *Lutje Hulsik et al., 2013* | PMID:23505368 | See Materials and methods |
| Antibody (llama nanobody) | α-gp41, 2H10-F | This study | | See Materials and methods |
| Antibody (llama nanobody) | α-gp41, 2H10-RKRF | This study | | See Materials and methods |
| Chemical compound, drug | Chelating sepharose FF | GE Healthcare | Cat# 17057501 | |
| Chemical compound, drug | Q sepharose FF | GE Healthcare | Cat# 17051010 | |
| Chemical compound, drug | Q sepharose FF | GE Healthcare | Cat# 17051010 | |
| Chemical compound, drug | n-Octyl-β-D-glucosid n-Octyl-β-D-Glucopyranoside | Anatrace | Cat#O311 | |
| Chemical compound, drug | CHAPS (3-[(3-cholamidopropyl) diméthylammonio]—1-propanesulfonate) | Euromedex | Cat#1083E | |
| Chemical compound, drug | 1-palmitoyl-2-oleoylphosphatid ylcholine (POPC) | Avanti Polar Lipids | Cat#850457P | |
| Chemical compound, drug | 1-palmitoyl-2-oleoyl-*sn*-glycero-3-phosphoethanolamine (POPE) | Avanti Polar Lipids | Cat#850757P | |
| Chemical compound, drug | 1-palmitoyl-2-oleoyl-*sn*-glycero-3-phospho-L-serine (POPS) | Avanti Polar Lipids | Cat#840034C | |
| Chemical compound, drug | sphingomyelin | Avanti Polar Lipids | Cat#860062C | |
| Chemical compound, drug | cholesterol | Avanti Polar Lipids | Cat#700000P | |
| Chemical compound, drug | N-(7-nitro-benz-2-oxa-1,3-diazol-4-yl) phosphatid ylethanolamine (N-NBD-PE) | Molecular Probes | Cat#N360 | |
| Chemical compound, drug | N-(lissamine Rhodamine B sulfonyl) phosphatid ylethanolamine (N-Rh-PE) | Molecular Probes | Cat#L1392 | |

*Continued on next page*

*Continued*

| Reagent type (species) or resource | Designation | Source or reference | Identifiers | Additional information |
|---|---|---|---|---|
| Chemical compound, drug | Biscinchoninic Acid microassay | Pierce | Cat# 23235 | |
| Chemical compound, drug | Bio-Rad Protein Assay Dye Reagent Concentrate | Biorad | Cat# 5000006 | |
| Software, algorithm | XDS | *Kabsch, 2010* | PMID:20124693 | |
| Software, algorithm | Phaser | *McCoy et al., 2007* | PMID:19461840 | |
| Software, algorithm | COOT | *Emsley et al., 2010* | PMID:20383002 | |
| Software, algorithm | PHENIX | *Adams et al., 2010* | PMID:20124702 | |
| Software, algorithm | SBGrid | *Morin et al., 2013* | https://sbgrid.org/ | |
| Software, algorithm | Pymol | Warren DeLano | http://www.pymol.org | |
| Software, algorithm | Aimless | *Evans and Murshudov, 2013* | PMID:23793146 | |
| Software, algorithm | STARANISO | *Tickle et al., 2018* | http://staraniso.globalphasing.org/cgi-bin/staraniso.cgi | |
| Software, algorithm | CHARMM-GUI | *Jo et al., 2008* | http://www.charmm-gui.org | |
| Software, algorithm | NAMD (Version 2.13) | *Phillips et al., 2005* | https://www.ks.uiuc.edu/Research/namd/ | |
| Software, algorithm | Prism 8 | GraphPad | https://www.graphpad.com/scientific-software/prism/ | |
| Software, algorithm | ForteBio analysis software version 11.1.0.25 | ForteBio | https://www.fortebio.com | |
| Software, algorithm | MicroCal Origin software (origin 7) | Malvern Panalytical (MicroCal) | https://www.malvernpanalytical.com | |
| Peptide, recombinant protein | NEQELLELDKWASLW NWFNITNWLWYIK (N-MPER) | This study | | See Materials and methods |
| Peptide, recombinant protein | *KKK*-NWFDITNWLWYI KLFIMIVGGLV-*KK* (C-MPER), | This study | | See Materials and methods |
| Commercial assay or kit | Biscinchoninic Acid microassay | Pierce | Cat# A53225 | |
| Commercial assay or kit | Bright-Glo Luciferase Assay System Streptavidin (SA) biosensors | Promega | Cat# E2610 | |
| Commercial assay or kit | Streptavidin (SA) biosensors | ForteBio | Cat#18–5019 | |
| Commercial assay or kit | EZ-LinkNHS-PEG4-Biotinylation Kit | ThermoFisher | Cat#21455 | |

## Cell lines

HEK 293 T cells were obtained from the American Type Culture Collection (ATCC CRL-11268).

TZM-bl cells were obtained through the NIH HIV Reagent Program, Division of AIDS, NIAID, NIH: TZM-bl Cells, ARP-8129, contributed by Dr. John C. Kappes and Dr. Xiaoyun Wu. HEK 293T and TZM-bl cells were maintained in Dulbecco's modified Eagle's medium supplemented with 10% heat-inactivated fetal bovine serum, 100 units/ml of Penicillin and 0.1 mg/ml of Streptomycin (all medium components from Gibco, Thermo Fisher Scientific) at 37°C, 5% $CO_2$, and 80% relative humidity. Both cell lines are regularly checked for absence of Mycoplasma (Mycoplasmacheck test by Eurofins Genomics (formerly GATC), Germany). No specific cell line authentication was performed.

## HIV-1 primary viruses

Env-pseudotyped viruses were prepared by co-transfection of HEK 293 T cells with plasmids encoding the respective *env* genes and the luciferase reporter HIV vector pNLluc-AM as described (*Rusert et al., 2009*). A full list of Env pseudotyped viruses generated with corresponding gene bank entry, subtype and Tier information is provided in *Table 1—source data 1*.

## GP41 expression and purification

DNA fragments encoding HIV-1 Env glycoprotein amino acids 512–594 (N-terminal chain, chain N) and residues 629 to 716 (C-terminal chain, chain C) were cloned into vectors pETM20 and pETM11 (PEPcore facility-EMBL), respectively. Chain N contains an N-terminal Flag-tag (DYKDDDDK sequence) and chain C an N-terminal enterokinase cleavage site (DDDDK) and two C-terminal arginine residues (*Figure 1A*). The sequence of chain N is GAMDYKDDDDK-512-AVGIGALLLGFLGAAG STMGAASMTLTVQARQLLSGIVQQQNNLLRAIEAQQHLLQLTVWGIKQLQARILAVERYLKDQQLLG-594 and of chain C is GAMDDDDK-629-MEWDREINNYTSLIHSLIEESQNQQE<u>KNEQELLELDKWA SLW**NWFNITNWLWYIKLFIMIVGGLVG**</u>LRIVFAVLSVVNRVRQGYSPLS-716-RR after TEV protease cleavage of the constructs. The N-MPER peptide region used in the fusion assay is underlined and the C-MPER peptide region is shown in bold.

Proteins were expressed separately in *E. coli* strain C41(DE3)(Lucigen). Bacteria were grown at 37°C to an $OD_{600nm}$ of 0,9. Cultures were induced with 1 mM IPTG at 37°C for 3 hr for gp41 chain N and at 25°C for 20 hr for gp41 chain C. Cells were lysed separately by sonication in buffer A containing 20 mM Tris pH 8, 100 mM NaCl and 1% CHAPS (3-[(3-cholamidopropyl) diméthylammonio]−1-propanesulfonate (Euromedex)). The supernatant was cleared by centrifugation at 53,000 g for 30 min. Gp41 chain N supernatant was loaded on a $Ni^{2+}$-sepharose column, washed successively with buffer A containing 1M NaCl and 1M KCl, then buffer A containing 50 mM imidazole. Gp41 chain N was eluted in buffer A containing 500 mM imidazole and further purified on a Mono Q column in buffer A and eluted with buffer B (buffer A with 0.5 M NaCl). Gp41 chain C was purified employing the same protocol as for gp41 chain N. Gp41 chain C was subsequently cleaved with TEV (Tobacco Etch Virus) protease for 2 hr at 20°C and then overnight at 4°C at a gp41 chain C:TEV ratio of 5:1 (w:w). After buffer exchange with a mono Q column using buffer A and elution in buffer B, uncleaved gp41 chain C was removed by a second $Ni^{2+}$-sepharose column in buffer A. TEV-cleaved gp41 chain C and un-cleaved gp41 chain N were then mixed in a molar ratio 4:1 and incubated overnight. To remove the excess of gp41 chain C, the gp41 complex was further purified on a 3rd $Ni^{2+}$-sepharose column in buffer A, washed with buffer A containing 50 mM imidazole and eluted with buffer A containing 500 mM imidazole. Subsequently the gp41 chain N TrxA-His-tag was removed by TEV digestion for 2 hr at 20°C and overnight at 4°C at a gp41:TEV ratio of 5:1 (w:w). Followed by another buffer exchange with a mono Q column (buffers A and B), uncleaved complex and the TrxA-His-tag fusion were removed by a 4th $Ni^{2+}$-sepharose column. The final gp41FP-TM complex was concentrated and loaded onto a Superdex 200 size exclusion column (SEC) in buffer C containing 20 mM Tris pH 8.0, 100 mM NaCl and 1% n-octyl β-D-glucopyranoside (Anatrace).

## Nanobody 2H10 expression

2H10 encoding DNA was cloned into the vector pAX51 (*Lutje Hulsik et al., 2013*) and expressed in the *E. coli* BL21(DE3) strain (ThermoFisher). Bacteria were grown at 37°C to an $OD_{600nm}$ of 0.7 and induced with 1 mM IPTG at 20°C for 20 hr. After harvesting by centrifugation, bacteria were resuspended in lysis buffer containing 20 mM Hepes pH 7.5 and 100 mM NaCl. Bacteria were lysed by

sonication and centrifuged at 48,000 g for 30 min. Cleared supernatant was loaded onto a Protein A sepharose column, washed with lysis buffer and eluted with 0.1 M glycine pH 2.9. Eluted fractions were immediately mixed with 1/5 vol of 1M Tris pH 9.0. 2H10 was then further purified by SEC on a superdex 75 column in PBS buffer. Genes corresponding to mutants of 2H10, 2H10-F (S100d) and 2H10-RKRF (S27R, S30K, S74R, and S100d) were synthesized (Biomatik) and the expressed mutant proteins were purified as described for the wild type. The 2H10 bi-head was purified as described (*Lutje Hulsik et al., 2013*).

## Circular dichroism

CD measurements were performed using a JASCO Spectropolarimeter equipped with a thermoelectric temperature controller. Spectra of gp41-TM were recorded at 20°C in 1 nm steps from 190 to 260 nm in a buffer containing PBS supplemented with 1% n-octyl β-D-glucopyranoside. For thermal denaturation experiments, the ellipticity was recorded at 222 nm with 1°C steps from 20° to 95°C with an increment of 80°C h$^{-1}$, and an averaging time of 30 s/step. For data analysis, raw ellipticity values recorded at 222 nm were converted to mean residue ellipticity.

## Isothermal titration calorimetry (ITC)

The stoichiometry and binding constants of 2H10 binding to gp41 FP-TM was measured by ITC200 (MicroCal Inc). All samples used in the ITC experiments were purified by SEC in a buffer containing 20 mM Tris pH 8.0, 100 mM NaCl and 1% n-octyl β-D glucopyranoside and used without further concentration. Samples and were equilibrated at 25°C before the start of the experiment. The ITC measurements were performed at 25°C by making 20 2 µl injections of 267 µM 2H10 to 0.2 ml of 19.5 µM gp41FP-TM. Curve fitting was performed with MicroCal Origin software. Three experiments were performed, with an average stoichiometry N = 1.1 +/- 0.2. 2H10 binds to gp41FP-TM with a KD of 2.1 µM ± 0.9.

## Bio-layer interferometry binding analysis

Binding measurements between antibodies (10E8 IgG, LN01 IgG, and 2H10) were carried out on an Octet Red instrument (ForteBio). For the determination of the binding between antibodies and gp41FP-TM, 10E8 IgG or LN01 IgG or 2H10 were labeled with biotin (EZ-Link NHS-PEG4-Biotin) and bound to Streptavidin (SA) biosensors (ForteBio). The biosensors loaded with the antibodies were equilibrated in the kinetic buffer (20 mM Tris pH 8.0, 100 mM NaCl and 1% n-octyl β-D glucopyranoside) for 200–500 s prior to measuring association with different concentrations of gp41FP-TM for 100–200 s at 25°C. Data were analyzed using the ForteBio analysis software version 11.1.0.25 (ForteBio). For 10E8, the kinetic parameters were calculated using a global fit 1:1 model and 2:1 model. For the determination of the binding of LN01 IgG and 2H10, KDs were estimated by steady state analysis. All bio-layer interferometry experiments were conducted at least three times.

## Immunoprecipitation of gp41FP-TM by bnAbs 10E8 and LN01

A total of 220 µg of Gp41FP-TM were incubated alone or with 50 µg of 10E8 or LN01 antibodies for 10 hr at 20°C in buffer C. The complex was loaded on Protein A sepharose affinity resin and incubated for 1 hr. The resin was subsequently washed three times with buffer C and eluted with SDS gel loading buffer and boiling at 95°C for 5 min. Samples were separated on a 15% SDS-PAGE and stained with Coomassie brilliant blue.

## Neutralization assay

The neutralization activity of the 2H10 variants and mAbs (IgG) 2F5 (*Muster et al., 1993*) and VRC01 (*Zhou et al., 2010*) was evaluated using TZM-bl cells and Env pseudotyped viruses as described (*Rusert et al., 2009*). Briefly, serial dilutions of inhibitor were prepared in cell culture medium (DMEM with 10% heat-inactivated FBS, 100 U/ml penicillin and 100 µg/ml streptomycin (all from Gibco)) and added at a 1:1 vol ratio to pseudovirus suspension in 384 well plates (aiming for 500,000–5,000,000 relative light units (RLU) per well in the absence of inhibitors). After one-hour incubation at 37°C, 30 µl of virus-inhibitor mixture was transferred to TZM-bl cells in 384 well plates (6000 cells/well in 30 µl cell culture medium supplemented with 20 µg/ml DEAE-Dextran seeded the previous day). The plates were further incubated for 48 hr at 37°C before readout of luciferase

reporter gene expression on a Perkin Elmer EnVision Multilabel Reader using the Bright-Glo Luciferase Assay System (Promega).

The inhibitor concentration (referring to the mix with cells, virus, and inhibitor) causing 50% reduction in luciferase signal with respect to a reference well without inhibitor (inhibitory concentration IC50) was calculated by fitting a non-linear regression curve (variable slope) to data from two independent experiments using Prism (GraphPad Software). If 50% inhibition was not achieved at the highest inhibitor concentration tested, a greater than value was recorded. To control for unspecific effects, all inhibitors were tested for activity against MuLV envelope pseudotyped virus.

## 2H10 membrane interaction

Lipids, 1-palmitoyl-2-oleoylphosphatidylcholine (POPC), 1-palmitoyl-2-oleoyl-*sn*-glycero-3-phosphoethanolamine (POPE), 1-palmitoyl-2-oleoyl-*sn*-glycero-3-phospho-L-serine (POPS), Sphingomyelin, and cholesterol (Avanti Polar Lipids) were dissolved in chloroform and mixed in a 1.2:2.6:1.0:2.4:5.5 ratio (w/w) as reported previously (*Chen et al., 2014*) and close to the HIV-1 envelope lipid composition (*Brügger et al., 2006*). After solvent evaporation, the lipid film was further dried under vacuum, and resuspended in PBS at a final concentration of 1 mg/ml. The liposome suspension was extruded through a 0.2 µm polycarbonate filter membrane. Fifty µg of 2H10, 50 µg of 2H10-RKRF and 100 µg 10E8 were incubated at room temperature without and with 50 µl of liposomes for 1.5 hr. The total volume of proteoliposome mixtures of 75 µl was then mixed with 150 µl of 70% sucrose generating the bottom fraction of the gradient, which was successively overlaid by 100 µl of 25% sucrose, 10% sucrose, and 5% sucrose PBS solutions. After centrifugation at 40,000 rpm in a Beckman Optima TL Ultracentrifuge using a SW55TI rotor for 3.5 hr at 10°C, four 75 µl fractions and the bottom input fraction (225 µl) were collected, separated by SDS-PAGE and protein bands were detected by Instant Blue staining.

## Fusion assay

The peptides used in the fusion inhibition experiments, NEQELLELDKWASLW NWFNITNWLWYIK (N-MPER) and *KKK*-NWFDITNWLWYIKLFIMIVGGLV-*KK* (C-MPER), were synthesized in C-terminal carboxamide form by solid-phase methods using Fmoc chemistry, purified by reverse phase HPLC, and characterized by matrix-assisted time-of-flight (MALDI-TOF) mass spectrometry (purity >95%). Peptides were routinely dissolved in dimethylsulfoxide (DMSO, spectroscopy grade) and their concentration determined by the Bicinchoninic Acid microassay (Pierce, Rockford, IL, USA).

Large unilamellar vesicles (LUV) were prepared following the extrusion method of *Hope et al., 1985*. 1-palmitoyl-2-oleoylphosphatidylcholine (POPC) and cholesterol (Chol) (Avanti Polar Lipids, Birmingham, AL, USA) were mixed in chloroform at a 2:1 mol:mol ratio and dried under a $N_2$ stream. Traces of organic solvent were removed by vacuum pumping. Subsequently, the dried lipid films were dispersed in 5 mM Hepes and 100 mM NaCl (pH 7.4) buffer, and subjected to 10 freeze-thaw cycles prior to extrusion 10 times through two stacked polycarbonate membranes (Nuclepore, Inc, Pleasanton, CA, USA). Lipid mixing with fusion-committed vesicles was monitored based on the resonance energy transfer assay described by *Struck et al., 1981*, with the modifications introduced by *Apellániz et al., 2014b*. The assay is based on the dilution of co-mixed N-(7-nitro-benz-2-oxa-1,3-diazol-4-yl)phosphatidylethanolamine (N-NBD-PE) and N-(lissamine Rhodamine B sulfonyl) phosphatidylethanolamine (N-Rh-PE) (Molecular Probes, Eugene, OR, USA), whereby dilution due to membrane mixing results in increased N-NBD-PE fluorescence. Vesicles containing 0.6 mol % of each probe (target vesicles) were added at 1:9 ratio to unlabeled vesicles (MPER peptide-primed vesicles). The final lipid concentration in the mixture was 100 µM. The increase in NBD emission upon mixing of target-labeled and primed-unlabeled lipid bilayers was monitored at 530 nm with the excitation wavelength set at 465 nm. A cutoff filter at 515 nm was used between the sample and the emission monochromator to avoid scattering interferences. The fluorescence scale was calibrated such that the zero level corresponded to the initial residual fluorescence of the labeled vesicles and the 100% value to complete mixing of all the lipids in the system (i.e. the fluorescence intensity of vesicles containing 0.06 mol % of each probe). Fusion inhibition was performed with bi-2H10, 2H10-RKRF and 2F5 Fabs at concentrations of 10 µg/ml and 20 µg/ml as indicated. Fabs 2F5 and 10E8 were used as positive controls in experiments of Ab-induced lipid-mixing inhibition.

Potencies of bi-2H10 and 2H10-RKRF Abs were compared in dose-response assays performed at concentrations in the 0–20 µg/ml range.

## Crystallization, data collection, and structure determination

For crystallization, 1 mg of gp41FP-TM was mixed with 1.5 mg of 2H10. The complex was purified by SEC on a Superdex 200 column in a buffer containing 100 mM NaCl, 20 mM Tris pH 8,0 and 1% n-octyl β-D-glucopyranoside and concentrated to 7–10 mg/ml. Crystal screening was performed at the EMBL High-Throughput Crystallization Laboratory (HTX lab, Grenoble) in 96-well sitting drop vapor diffusion plates (Greiner). Following manual refinement of crystallization conditions, crystals of gp41FP-TM in complex with 2H10 were grown by mixing 1 µl of protein with 1 µl of reservoir buffer containing 0.1 M sodium citrate pH 6.0, 0.2 M ammonium sulfate, 20% polyethylene glycol 2000 and 0.1 M NaCl at 20°C (293 K) in hanging drop vapor diffusion plates. Before data collection, crystals were flash frozen at 100K in reservoir solution supplemented with 1% n-octyl β-D-glucopyranoside and 25% ethylene glycol for cryo-protection.

Data were collected on the ESRF beamline ID30b at a wavelength of 0.9730 Å. Data were processed with the program XDS (*Kabsch, 2010*). The data from two crystals were merged with Aimless (*Evans and Murshudov, 2013*). The data set displayed strong anisotropy in its diffraction limits and was submitted to the STARANISO Web server (*Tickle et al., 2018*). The merged STARANISO protocol produced a best-resolution limit of 3.28 Å and a worst-resolution limit of 4.74 Å at a cutoff level of 1.2 for the local $I_{mean}$ / $\sigma(I_{mean})$ (note that STARANISO does not employ ellipsoidal truncations coincident with the crystal axes). The gp41FP-TM-2H10 crystals belong to space group C 2 2 $2_1$ and the structure was solved by molecular replacement using the program Phaser (*McCoy et al., 2007*) and pdb entries 1env and 4b50. The model was rebuilt using COOT (*Emsley et al., 2010*) and refined using Phenix (*Adams et al., 2010*). Data up to 3.28 Å were initially used for model building but were finally truncated to 3.8 Å. Statistics for data reduction and structure refinement are presented in *Table 1*.

One copy of gp41FP-TM in complex with 2H10 are present in the asymmetric unit. Numbering of the nanobody 2H10 was performed according to Kabat. The gp41FP-TM-2H10 complex was refined to 3.8 Å data with an R/Rfree of 26.6/30.8%. 99.6% of the residues are within the most favored and allowed regions of a Ramachandran plot (*Evans and Murshudov, 2013*). Some of the crystallographic software used were compiled by SBGrid (*Morin et al., 2013*). Atomic coordinates and structure factors of the reported crystal structures have been deposited in the Protein Data Bank (https://www.rcsb.org; PDB: 7AEJ).

## Figure generation

Molecular graphics figures were generated with PyMOL (W. Delano; The PyMOL Molecular Graphics System, Version 1.8 Schrödinger, LLC, http://www.pymol.org). The symmetry axis shown in *Figure 1C* was calculated with the program Galaxy/SymD using the core of gp41 HR1 residues Ala541 to Leu587 (*Tai et al., 2014*).

## Molecular dynamics (MD) simulation

### Molecular assays

Starting from the crystal structure determined herein, we built two molecular assays. The first is based on the crystal structure of gp41FP-TMin complex with 2H10. The disordered FP and TM residues were modeled as helices (chain A FP 512–526 and TM 694–707, chain B FP 512–517 and TM 701–707, chain C FP 512–517). The second assay is based on a symmetric gp41 model which was generated from protomer B by applying a three-fold symmetry operation. Missing FP residues 512–517 and TM residues 701–709 were modeled as helices (*Figure 4—figure supplement 1B*), and TM residues were modeled based on Env gp41 TM structures (pdb entry 6SNE and 6B3U). All residues were taken in their standard protonation state.

The first assay included a fully hydrated membrane composed of 190 cholesterol, 40 1-palmitoyl-2-oleoyl-glycero-3-phosphocholine (POPC), 88 1-palmitoyl-2-oleoyl-sn-glycero-3-phospho-ethanol-amine (POPE), 36 1-palmitoyl-2-oleoyl-sn-glycero-3-phospho-L-serine (POPS) and 56 N-stearoyl sphingomyelin, present in the HIV-1 lipid envelope (*Brügger et al., 2006*), using the CHARMM-GUI interface (*Jo et al., 2008*; *Wu et al., 2014*). The resulting molecular assembly consisted of about

178,000 atoms in a rhomboidal cell of $106 \times 106 \times 169$ Å$^3$. The second computational assay featured a water bath of $91 \times 91 \times 114$ Å$^3$, representing a total of 95,700 atoms. Both assays were electrically neutral, with a NaCl concentration set to 150 mM.

## Molecular dynamics

All simulations were performed using the NAMD 2.14 program (*Phillips et al., 2005*). Proteins, cholesterol, lipids and ions were described using the CHARMM force field (*Best et al., 2012*; *Klauda et al., 2010*; *MacKerell et al., 1998*) and the TIP3P model (*Jorgensen et al., 1983*) was used for water. MD trajectories were generated in the isobaric-isothermal ensemble at a temperature of 300 K and a pressure of 1 atm. Pressure and temperature were kept constant using the Langevin thermostat and the Langevin piston method (*Feller et al., 1995*), respectively. Long-range electrostatic interactions were evaluated by the particle-mesh Ewald (PME) algorithm (*Darden et al., 1993*). Hydrogen mass repartitioning (*Hopkins et al., 2015*) was employed for all simulations, allowing for using a time step of 4 fs. Integration was performed with a time step of 8 and 4 fs for long- and short-range interactions, respectively, employing the r-RESPA multiple time-stepping algorithm (*Tuckerman et al., 1992*). The SHAKE/RATTLE (*Andersen, 1983*; *Ryckaert et al., 1977*) was used to constrain covalent bonds involving hydrogen atoms to their experimental lengths, and the SETTLE algorithm (*Miyamoto and Kollman, 1992*) was utilized for water.

The computational assay formed by gp41 in an aqueous environment was simulated for a period of 1 µs, following a thermalization of 40 ns. For the gp41FP-TM/2H10 complex, the lipid bilayer was first thermalized during 200 ns using soft harmonic restraints on every dihedral angle of the protein backbones, allowing the complex to align optimally with its membrane environment. Following the equilibration step, a production run of 1 µs was performed.

The final structural model of hydrated gp41 was also embedded in the HIV-1-like envelope membrane employed for the gp41FP-TM/2H10 crystal structure complex. The same 200 ns equilibration protocol was applied followed by a production run of 1 µs.

## Acknowledgements

WW acknowledges support from the Institut Universitaire de France (IUF), from the European Union's Horizon 2020 research and innovation program under grant agreement No. 681137, H2020 EAVI and the platforms of the Grenoble Instruct-ERIC center (IBS and ISBG; UMS 3518 CNRS-CEA-UGA-EMBL) within the Grenoble Partnership for Structural Biology (PSB). Platform access was supported by FRISBI (ANR-10-INBS-05–02) and GRAL, a project of the University Grenoble Alpes graduate school (Ecoles Universitaires de Recherche) CBH-EUR-GS (ANR-17-EURE-0003). IBS is part of the CEA DRF Interdisciplinary Research Institute of Grenoble (IRIG). JLN acknowledges funding from Spanish MCIU (RTI2018-095624-B-C21; MCIU/AEI/FEDER, UE) and Basque Government (IT1196-19). We thank Miriam Hock and Serafima Guseva for previous contributions to the project, Davide Corti for providing bnAb LN01, the ESRF-EMBL Joint Structural Biology Group for access and support at the ESRF beam lines, J Marquez (EMBL) from the HTX crystallization facility and C Mas and J-B Reiser for assistance on ISBG platforms.

# Additional information

## Funding

| Funder | Grant reference number | Author |
| --- | --- | --- |
| H2020 Health | 681137 | Winfried Weissenhorn |
| Agence Nationale de la Recherche | ANR-17-EURE-0003 | Winfried Weissenhorn |
| Ministerio de Economía, Industria y Competitividad, Gobierno de España | BIO2015-64421-R | Jose L Nieva |
| Ministerio de Ciencia y Tecnología | RTI2018-095624-B-C21 | Jose L Nieva |

| French Infrastructure for Integrated Structural Biology | ANR-10-INBS-05–02 | Winfried Weissenhorn |

The funders had no role in study design, data collection and interpretation, or the decision to submit the work for publication.

### Author contributions

Christophe Caillat, Christophe J Chipot, Conceptualization, Formal analysis, Investigation, Visualization, Methodology, Writing - original draft; Delphine Guilligay, Nikolas Friedrich, Investigation, Visualization, Methodology, Writing - original draft; Johana Torralba, Investigation, Visualization; Jose L Nieva, Conceptualization, Supervision, Funding acquisition, Methodology, Writing - original draft; Alexandra Trkola, Conceptualization, Supervision, Funding acquisition; François L Dehez, Conceptualization, Formal analysis, Supervision, Funding acquisition, Investigation, Visualization, Methodology, Writing - original draft; Winfried Weissenhorn, Conceptualization, Formal analysis, Supervision, Funding acquisition, Methodology, Writing - original draft, Project administration, Writing - review and editing

### Author ORCIDs

Christophe Caillat https://orcid.org/0000-0003-2959-1871
Nikolas Friedrich http://orcid.org/0000-0003-0694-657X
Alexandra Trkola http://orcid.org/0000-0003-1013-876X
Christophe J Chipot http://orcid.org/0000-0002-9122-1698
Winfried Weissenhorn https://orcid.org/0000-0001-5532-4959

### Decision letter and Author response

Decision letter https://doi.org/10.7554/eLife.65005.sa1
Author response https://doi.org/10.7554/eLife.65005.sa2

## Additional files

### Supplementary files

• Transparent reporting form

### Data availability

Diffraction data have been deposited in PDB under the accession code 7AEJ. All data generated or analysed during this study are included in the manuscript and supporting files. Source data files have been provided for Table 2.

The following dataset was generated:

| Author(s) | Year | Dataset title | Dataset URL | Database and Identifier |
|---|---|---|---|---|
| Caillat C, Guilligay D, Weissenhorn W | 2020 | Crystal structure of asymmetric HIV-1 gp41 containing all membrane anchors | https://www.rcsb.org/structure/7AEJ | RCSB Protein Data Bank, 7AEJ |

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
