## [Decision Letter]

**Acceptance summary:**

This manuscript describes an X-ray structure at 3.8 A resolution of the gp41 subunit of the HIV envelope protein complex (Env) using a construct that spans the membrane-interacting regions, the fusion peptide (FP) and the trans-membrane (TM) region, respectively located at its N- and C-termini and for which no structural information was available. The structure reveals an asymmetric trimer with the core portion folded as in previous structures, but with the N- and C-terminal ends diverging from 3-fold symmetry. The regions visible in the present structure are important functional elements, and in this context the paper provides new insights into the function of gp41.

**Decision letter after peer review:**

Thank you for submitting your article "Structure of HIV-1 gp41 with its membrane anchors targeted by neutralizing antibodies" for consideration by *eLife*. Your article has been reviewed by 3 peer reviewers, and the evaluation has been overseen by Axel Brunger as Reviewing Editor and John Kuriyan as the Senior Editor. The reviewers have opted to remain anonymous.

The reviewers have discussed the reviews with one another and the Reviewing Editor has drafted this decision to help you prepare a revised submission.

We would like to draw your attention to changes in our policy on revisions we have made in response to COVID-19 (https://elifesciences.org/articles/57162).

Summary:

This manuscript describes an X-ray structure of the gp41 subunit of the HIV envelope protein complex (Env) using a construct that spans the membrane-interacting regions, the fusion peptide (FP) and the trans-membrane (TM) region, respectively located at its N- and C-termini and for which no structural information was available. The post-fusion form of gp41 is a trimer of hairpins with a 6-helix bundle core constituted by a central parallel α-helical coiled coil made by an N-terminal α helix, spanning a region termed "heptad repeat 1" (HR1), and an antiparallel C-terminal helix running along the grooves of the coiled coil, spanning a region termed HR2. In this work, to obtain crystals the authors produced two fragments of gp41 that include the HR1 and HR2 regions respectively, an N-terminal fragment with roughly the N-terminal half (70 residues), and a C-terminal fragment of ∼90 residues. The cytosolic tail of gp41, spanning the very C-terminal 150 residues, is absent in the construct. To obtain crystals, they produced the two fragments separately and mixed them to obtain the trimeric complex, termed gp41FP-TM, which crystallized in the presence of a single-chain camelid antibody, 2H10, which binds at the MPPR to make a complex at a stoichiometry 1:3 (i.e, one 2H10 per gp41FP-TM trimer). The resulting structure at 3.8 Å resolution shows an asymmetric trimer with the core portion folded as in previous structures, but with the N- and C-terminal ends diverging from 3-fold symmetry. The regions visible in the present structure are important functional elements, and in this context the paper provides new insights into the function of gp41. Also, the model of the potential interactions of the resulting complex with membranes is validated by mutagenesis of the 2H10 nanobody, which increases in functionality when modifying residues predicted to be directly in contact with the membrane (either by making them more hydrophobic, or positively charged in order to interact ionically with the phosphate head groups of the lipids). The paper also reports a fusion assay involving pre-activated liposomes with peptides spanning the MPER, to show that the antibody blocks fusion, and the antibody mutants with higher affinity (including a bivalent version of 2H10), do so more efficiently. However, the molecular dynamics simulations and models are highly speculative.

Essential revisions:

The overall resolution of the crystal structure is 3.8 Å with a data completeness of 78.01%, but the highest resolution shell has a completeness of 54.69% and a *R_merge_* of 150.8% (after STARANISO truncation, *R_merge_* in the resolution shell 3.97 Å – 3.85 Å is 78.7%). It is hard to tell the quality of the final map used for model building, in particular, for the fusion peptides and transmembrane segments. How much do the noisy high resolution reflections contribute to the refinement and map calculation? In addition, at this resolution, is there enough side chain density for one to be confident about the sequence register? In any case, it might be useful to show some representative density figures and B factors of the final model in the supplemental material to give reader a better sense how reliable the various regions of the model really are.

In Figure 1—figure supplement 1 and Figure 1—figure supplement 2, it is a bit odd that the larger chain C peptide looks smeary, but the smaller chain N is much sharper (sometimes looks like more than one band). Have these peptides been checked by mass spec? Since it is difficult to read the protein sequence from a 3.8 Å map, it would be important to confirm that the heterogeneity of the ends in density was not coming for the samples.

It is unclear whether the asymmetrical structure of gp41 could accommodate the bivalent 2H10 (bi-2H10) or not. If not, the authors might want to speculate why bi-2H10 is much more potent than the 2H10 nanobody.

p.5, Ln 58 – "packing effects… can be excluded'. While crystal contacts may not readily explain the asymmetric conformation of the FP and TM domains, it is possible that the observed conformation may have been stabilized by (i) the BOG detergent. Also would the symmetric 6HB conformation clash with the crystal lattice, and if so, the crystal packing does rule out a symmetric conformation? Please comment.

L132. Related to the previous point, it is not obvious that it is the antibody that induces the bent conformation. It is more likely that the detergent-solubilized protein maintains an equilibrium between multiple conformations, and that the nanobody selects one of them. Such a sampling of conformations is in line with gp41's function, as it needs to sample asymmetric conformations, in particular of this membrane interacting regions, to drive membrane fusion. There is a longstanding controversy in biochemistry between induced fit and conformational selection, but there are ways to distinguish them (see, for instance, https://doi.org/10.1016/j.bpc.2014.03.003). The point is not that the authors explore this suggestion with their construct, but rather that they consider the terminology used, as the end effect is the same. Just saying that the antibody stabilized an asymmetric form of the protein would be enough, without saying that it "induces it".

Lines 171-173. The bent on chain A-C at the epitope appears to be stabilized by 2H10 binding (with the caveat mentioned above about induced fit/conformational selection). But chain A-C has the shortest ordered density for its TM helix in the crystals. And the actual orientation of the TM helices of the two other chains, which are longer, may be stabilized by crystal packing. The authors indeed point to packing interactions made by chain C-B. But the same Figure suggests that chain C-C, which has the longest ordered TM helix, appears to be stabilized in the observed conformation by packing against the TM helix of chain C-B. This raises the question of the biological significance of the actual docking that the authors make on membranes, which they say is validated by MD.

p.6 In the lipid mixing experiments, bi-2H10 shows more potent inhibition than 2F5 but 2F5 is more potent than bi-2H10 in Table 1. This suggests that the lipid mixing assay may not represent the HIV gp41 mediated fusion reaction. Please clarify.

Do the mutations in 2H10-F and 2H10-RKRF that enhance membrane interactions and neutralization also increase autoreactivity or polyreactivity (toward Hep2 cells, lipids, etc.)?

Lines 112 and around: the N and C chains were purified separately. Do they behave as monodisperse monomers? Or does the N chain make trimers and the C chain behaves as monomer? Was SEC done previous to mixing the two to assemble the mono-disperse complex displayed in Figure 1, supplement 1B?

Lines 114-115: please compare the thermostability of both constructs in parallel in the same experiment, to make this statement stronger.

L131. It is not clear that the TM region and FP in the three chains in the trimer point in opposite directions, as stated. Figure 1A appears to indicate that they all point toward the right of the Figure.

Lines 131-140: It is not mentioned in this description what surface area is buried by the nanobody in the complex, and whether or not all three CDRs are involved in the interaction. These data should be provided, even if at 3.8 Å resolution the atomic model may not be very accurate. Does it contact only chain C-A, or are there contacts with the other chains stabilizing the interaction? The authors should be more explicit here.

Lines 161-168. The relevance of the molecular dynamics is questionable since there are many assumptions being made. The MD simulations of the protein in a lipid bilayer mimicking the composition of the HIV envelope are described twice in the manuscript (here and then from lines 233 on). The overall flow seems a little strange, although the present paragraph is to anticipate that the CDR3 of 2H10 is predicted to also interact with the membrane. In the Methods section, it is explained that to run the MD calculations the regions that were not visible were completed as α-helical extensions. But apparently only for FP residues 512 to 518 and TM from residues 700 to 709, as stated in line 532 in the Methods section. But according to Figure 1C, chain A ends at residue 693, so this means that the segment 694 to 700 was also built for this chain? Just as an α helix extending from 693? And what about the FP for chain A, which according to Figure 1C only starts at position 527? Was the FP not used for this segment? It is stated in lines 155-158 that the FP may not be helical upstream of residue 518. At any rate, as modeled in Figure 1, Figure supplement 4A, the two FP helices modeled end at the middle of the membrane, in the aliphatic moiety. The charged N-termini for the two chains modeled will be very unstable there, so there is no wonder that the MD moves them to reach the phosphate headgroups of the lipids across the membrane. This model goes against the current paradigm that the fusion peptides only interact with the outer leaflet of the target membrane, and does not cross it. It is questionable how the model has been docked into the membrane. Figure 1 Figure supplement 4B indeed suggests that the two FP helices traverse the membrane to interact with the lipid headgroups of the inner leaflet. Please soften the statements regarding the molecular dynamics simulations and model and emphasize that it they are highly speculative.

Lines 233-266. Related to the previous point, the authors should compare by also docking the FP on the membrane not as in the symmetrized X-ray structure, which is derived from the protein having this flexible non-polar region within a detergent micelle, but with the FP anchored only on the outer leaflet, as it would initially insert into target membranes in an extended intermediate conformation. In this respect, the illustration of Figure 5D is highly mis-leading, as the peptides are not expected to insert into the target membrane in the proposed way, with the charged N-termini within the aliphatic moiety of the membrane. Figure 5 indeed shows that lipid head groups move to the center of the membrane in order to compensate for the charge at the N-termini, which is highly unlikely. This Figure should be adapted to reflect this.

Lines 220-222. Did the authors check that 2F5 does not bind the gp41FP-TM construct? It could select for a different conformation than does 2H10 and still bind. Or does it have to bind its epitope exclusively when in the context of pre-fusion gp160? Because this antibody was used in the fusion experiments, it should be possible to test its binding properties. And the reverse question: Do 10E8 or LN01 inhibit fusion in the system described in the previous paragraph? Since the peptide used should include their epitopes, they could also have a fusion blocking effect.

---

## [Author Response]

Essential revisions:The overall resolution of the crystal structure is 3.8 Å with a data completeness of 78.01%, but the highest resolution shell has a completeness of 54.69% and a R_merge_ of 150.8% (after STARANISO truncation, R_merge_ in the resolution shell 3.97 Å – 3.85 Å is 78.7%). It is hard to tell the quality of the final map used for model building, in particular, for the fusion peptides and transmembrane segments. How much do the noisy high resolution reflections contribute to the refinement and map calculation? In addition, at this resolution, is there enough side chain density for one to be confident about the sequence register? In any case, it might be useful to show some representative density figures and B factors of the final model in the supplemental material to give reader a better sense how reliable the various regions of the model really are.

Molecular replacement using the coordinates of the gp41 core (1env) and of 2H10 (4b50) resulted in only one solution placing the gp41 core unambiguously in the electron density. Datasets of several crystals have been analysed and the determination of the space/point group by XDS, pointless (and other software) always gave as solution the point group C 2 2 2. Space groups of lower symmetry have also been tested. Zanuda was used for refinement based checking of the space group (P1, P 1 21 1, C 1 2 1 and C 2 2 21). Changing the screw axis in C 2 2 21 was also tested. However, the only space group that gave a clear solution by molecular replacement was C 2 2 21 with a TFZ score of 39.9.

We added 2Fo-Fc composite omit maps of two close-up regions, the HR1-HR2 core and the kinked MPER conformation of chain C to show the sequence register in Figure 1—figure supplement 3. Notably, the same kinked MPER conformation was reported in the original 10E8-MPER peptide structure (4G6F). The transmembrane domains and FP regions were subsequently built in the 3.8 Å resolution map. Although, the map provides much less precision in the position of the atoms, we are confident in the chain traces. For example, chain B forms a continuous helix of HR2-MPER-TM and its structure is in agreement with NMR and crystal structures of MPER-TM (6SNE and 6B3U). Furthermore, chain B is in the same conformation as the protomers present in gp41 structures including MPER (3k9a) and FPPR-MPER (2x7r) as shown in Figure 1—figure supplement 4.

In Figure 1—figure supplement 1 and Figure 1—figure supplement 2, it is a bit odd that the larger chain C peptide looks smeary, but the smaller chain N is much sharper (sometimes looks like more than one band). Have these peptides been checked by mass spec? Since it is difficult to read the protein sequence from a 3.8 Å map, it would be important to confirm that the heterogeneity of the ends in density was not coming for the samples.

Chain C looks always smeary on SDS-PAGE. We have analyzed the complex by mass spectrometry to confirm the integrity of the chains, Author response image 1. The calculated MW of chain N is 10134.75 Da and of chain C is 11688.32 Da. The mass detected by MALDI-TOF for chain N is 10132 Da, and for chain C 11686 Da, in agreement with the calculated MW. We detect only 2/3 minor peaks, which might be degradation/contamination products and do not explain the abnormal SDS-PAGE migration of chain C.

**Author response image 1. sa2fig1:** MALDI-TOF analysis of gp41FP-TM. The raw spectrum shows the single and doubly charged ion of chains N and C.

It is unclear whether the asymmetrical structure of gp41 could accommodate the bivalent 2H10 (bi-2H10) or not. If not, the authors might want to speculate why bi-2H10 is much more potent than the 2H10 nanobody.

The extended 2H10 epitope is only present in one chain, indicating that a bi-head is unlikely to interact with the two chains within the same trimer in the conformation of the crystal structure. We have previously suggested that bi-2H10 is more potent in the neutralization assay (Lutje-Hulsik et al. 2013, PLoS Pathogens; doi: 10.1371/journal.ppat.1003202) because of the avidity effect by binding two epitopes at the same time. SPR measurement presented in Lutje-Hulsik et al. revealed binding to a fusion intermediate conformation of gp41 (gp41int) with a KD of 29 nM for wt 2H10 and 2.5 nM for bi-2H10. The optimal length of the linker of bi-2H10 is 15 amino acids, which would also allow binding to different trimers at the fusion site.

We have added the following sentence in the discussion:

“This, thus, confirmed 2H10 as a modest anti-MPER neutralizing antibody, whose neutralization potency is not only enhanced by using bi-2H10 with increased avidity (Lutje Hulsik et al., 2013), but also by increasing potential membrane interaction of the 2H10 monohead.”

p.5, Ln 58 – "packing effects… can be excluded'. While crystal contacts may not readily explain the asymmetric conformation of the FP and TM domains, it is possible that the observed conformation may have been stabilized by (i) the BOG detergent.

The asymmetric conformation of the membrane anchors are not explained by crystal packing (see our response below). We also think that its stabilization by β-OG is unlikely, since the T_M_ is slightly increased in the gp41FP-TM construct (in β-OG buffer) in comparison to gp41 without FP and TM (Buzon et al. PLoS Pathogen). This increase of the T_M_ is in agreement with the MD model of the post-fusion conformation (without 2H10), which reveals coiled-coil interactions within the trimeric FP, and lateral interactions between the FP and the TM domains.

Also would the symmetric 6HB conformation clash with the crystal lattice, and if so, the crystal packing does rule out a symmetric conformation? Please comment.

The electron density is in agreement with the asymmetric model presented here and superimposing the 6HB “core structure” would not generate clashes. However, a model of the 6HB core including MPER-TM and FP, such as the post-fusion model presented in Figure 4 would produce clashes between the C-terminal residues of the TM domains of two symmetry-related molecules.

L132. Related to the previous point, it is not obvious that it is the antibody that induces the bent conformation. It is more likely that the detergent-solubilized protein maintains an equilibrium between multiple conformations, and that the nanobody selects one of them. Such a sampling of conformations is in line with gp41's function, as it needs to sample asymmetric conformations, in particular of this membrane interacting regions, to drive membrane fusion.

We agree with the reviewers that 2H10 might sample one conformation out of an equilibrium between multiple conformations, although a thermostability of ~ 93°C is indicative of some limitations in the conformational flexibility.

There is a longstanding controversy in biochemistry between induced fit and conformational selection, but there are ways to distinguish them (see, for instance, https://doi.org/10.1016/j.bpc.2014.03.003). The point is not that the authors explore this suggestion with their construct, but rather that they consider the terminology used, as the end effect is the same. Just saying that the antibody stabilized an asymmetric form of the protein would be enough, without saying that it "induces it".

We agree with the reviewers, and we have changed this accordingly throughout the text.

Lines 171-173. The bent on chain A-C at the epitope appears to be stabilized by 2H10 binding (with the caveat mentioned above about induced fit/conformational selection). But chain A-C has the shortest ordered density for its TM helix in the crystals. And the actual orientation of the TM helices of the two other chains, which are longer, may be stabilized by crystal packing. The authors indeed point to packing interactions made by chain C-B. But the same Figure suggests that chain C-C, which has the longest ordered TM helix, appears to be stabilized in the observed conformation by packing against the TM helix of chain C-B. This raises the question of the biological significance of the actual docking that the authors make on membranes, which they say is validated by MD.

There are no defined crystal contacts stabilizing the TM orientations (Figure 1—figure supplement 6). We apologize for the orientation shown in the previous Figure supplement 5, suggesting a defined crystal contact between C-terminal chains of the B protomer.

Crystal packing is, thus, unusual and we suggest that the disordered regions of the C-termini, such as chain B residues 701 to 716 might stabilize interlayer contacts, which are less regular, and hence, not defined by electron density at this resolution. We acknowledge that this is quite unusual for crystal packing.

An error in space group setting can be excluded as out of all space group settings tested as stated above, only C2221 provided a clear solution.

p.6 In the lipid mixing experiments, bi-2H10 shows more potent inhibition than 2F5 but 2F5 is more potent than bi-2H10 in Table 1. This suggests that the lipid mixing assay may not represent the HIV gp41 mediated fusion reaction. Please clarify.

Antibody 2F5 was included as a positive control in virus neutralization and fusion blocking assays because the linear epitope recognized by 2H10 overlapped with that recognized by 2F5. However, we note that, whereas the IgG was used in the neutralization assays (Table 2), the Fab version of 2F5 was used in the fusion inhibition assay (Figure 2). Thus, we infer that avidity effects may account for the slightly better performance of the 2F5 IgG in comparison with bi-2H10 in neutralization, an effect not observed in fusion inhibition. We now specify more clearly the different positive controls used in both assays (see Materials and methods).

Regarding the relevance of the lipid-mixing assay for the gp41-induced fusion, we agree with the reviewers that peptide-induced fusion in vitro can only recapitulate certain aspects of the process. However, recognition of the MPER peptides inserted in the membrane and blocking of their restructuring activity is relevant for the mechanism of molecular recognition of MPER Abs for the following reasons:

First, the capacity for blocking the lipid-mixing process correlates with the neutralization potency of the MPER Abs (ref: Apellaniz et al., 2014b). To illustrate this point, please see also Author response image 2, which compares 10E8 Fab variants with regard to neutralization and lipid-mixing inhibition. Since both activities of 10E8 and its variants correlate, we believe that these observations warrant the use of the assay to compare different 2H10 variants.

**Author response image 2. sa2fig2:** Neutralization potency of 10E8 Fab variants (cell entry inhibition measured against HIV-1 virions pseudotyped with JR-CSF Env) and their ability to block C-MPER-induced lipid-mixing. The top diagram (Figure 2—figure supplement 1B) shows the mutated residues and their positions with respect to the putative membrane binding.

Second, MPER antibodies interact directly with membrane-bound peptide epitopes, change their insertion state (depth level and oligomerization state) and inhibit membrane restructuring (permeabilization and fusion).

Thus, to address this question, we extended the Results section accordingly – line 196:

“In the experimental setting (Apellaniz et al., 2014b) a vesicle population is primed for fusion by addition of the N-MPER peptide containing the 2H10 epitope, which produces a fluorescence intensity spark at time 20 s (Figure 2A). […] MPER antibodies have been shown to interact directly with membrane-bound peptide epitopes, thereby changing their insertion state (depth and oligomerization levels) and inhibiting their capacity to induce fusion…”.

Line 216:

“The use of the 10E8 Fab as an additional control further supports the blocking effect mediated by epitope recognition. […] Following this trend, fusion inhibition levels estimated as a function of the antibody concentration confirmed the apparent higher potency exhibited by the bi-2H10 (Figure 2D).”

Do the mutations in 2H10-F and 2H10-RKRF that enhance membrane interactions and neutralization also increase autoreactivity or polyreactivity (toward Hep2 cells, lipids, etc.)?

We have added new data presented in Figure 1—figure supplement 7. Using a liposome flotation assay, we show that 2H10 and 2H10-RKFR do not reveal detectable membrane interaction in vitro whereas bnAb 10E8 reveals some membrane binding in this assay.

Lines 112 and around: the N and C chains were purified separately. Do they behave as monodisperse monomers? Or does the N chain make trimers and the C chain behaves as monomer? Was SEC done previous to mixing the two to assemble the mono-disperse complex displayed in Figure 1, supplement 1B?

We described the assembly and purification protocol in the methods section. Both chains were purified separately. They were not separated by SEC prior to complex formation. We have purified chain-C prior to analyzing it by SEC, which showed a monodisperse peak eluting at 13.5 ml (between the elution peak of 150 kDa and 75 kDa marker proteins) from a superdex 200 SEC column (Figure 1, Lutje-Hulsik et al. PLoS Pathogens 2013). In an unrelated experiment, the chain-N fusion protein was separated by SEC, revealing a monodisperse peak eluting at 12 ml from a superdex 200 SEC column.

Lines 114-115: please compare the thermostability of both constructs in parallel in the same experiment, to make this statement stronger.

We agree with the reviewers that a side-by-side comparison could strengthen the conclusion. Since the original experiments have been performed, our JASCO CD spectrophotometer broke down and has not yet been replaced. Due to restrictions imposed by the ongoing health crisis, we did not have access to a comparable instrument in a near-by facility. We had, however, previously performed side-by side CD experiments with a shorter gp41FP-TM construct in a buffer containing DDM, which also showed a slight increase of the Tm compared to gp41 without FP and TM. See Author response image 3.

**Author response image 3. sa2fig3:** Gp41(512-711) used in this experiment, composed of residues 512-581 and 628-711 (including FP-FPPR and MPER-TM), reveals a Tm of 90°C (green curve, buffer DDM) and gp41(528-683) (including FPPR and MPER) has a Tm of 88°C (black curve, no detergent). Please note that gp41FP-TM used in the current study is composed of residues 512-594 with a 13 aa longer N-terminal coiled coil and a 5 residue C-terminal extension to residue 716. The two curves are shown in comparison to the gp41 core structure composed of residues 541-665 (blue curve, no detergent).

L131. It is not clear that the TM region and FP in the three chains in the trimer point in opposite directions, as stated. Figure 1A appears to indicate that they all point toward the right of the Figure.

We have changed “opposite directions” to “different directions”. We also added Figure 1C, which includes the trimer axis of the symmetric 6-helical bundle core structure, and demonstrates further that FP and TM point in different directions.

Lines 131-140: It is not mentioned in this description what surface area is buried by the nanobody in the complex, and whether or not all three CDRs are involved in the interaction. These data should be provided, even if at 3.8 Å resolution the atomic model may not be very accurate. Does it contact only chain C-A, or are there contacts with the other chains stabilizing the interaction? The authors should be more explicit here.

We have included the buried surface area and added on line 137:

“The protomer A chain C 2H10 epitope spans from residues Q658 to N671, which is involved in a series of polar contacts with 2H10 covering a total buried surface of 712.2 Å2.”

Lines 161-168. The relevance of the molecular dynamics is questionable since there are many assumptions being made. The MD simulations of the protein in a lipid bilayer mimicking the composition of the HIV envelope are described twice in the manuscript (here and then from lines 233 on). The overall flow seems a little strange, although the present paragraph is to anticipate that the CDR3 of 2H10 is predicted to also interact with the membrane. In the Methods section, it is explained that to run the MD calculations the regions that were not visible were completed as α-helical extensions. But apparently only for FP residues 512 to 518 and TM from residues 700 to 709, as stated in line 532 in the Methods section. But according to Figure 1C, chain A ends at residue 693, so this means that the segment 694 to 700 was also built for this chain? Just as an α helix extending from 693? And what about the FP for chain A, which according to Figure 1C only starts at position 527? Was the FP not used for this segment?

We used MD simulations for two different computational experiments. First, we used MD to analyze the crystal structure of the complex placed in a lipid environment with the aim to determine whether the membrane anchors, which are surrounded by detergent in the crystal structure are stable in a lipid environment. This is shown in Figure 1—figure supplement 5. The computational experiment was performed with the gp41FP-TM-2H10 crystal structure model, which has the missing FP and TM residues modeled as helices in all 3 protomers.

We clarified this in the Methods section line 594:

“Starting from the crystal structure determined herein, we built two molecular assays. The first is based on the crystal structure of gp41FP-TM in complex with 2H10. […] All residues were taken in their standard protonation state.”

It is stated in lines 155-158 that the FP may not be helical upstream of residue 518.

This is correct. The visible N-termini of N-terminal chains C and B start at residue 518. The remaining six residues are disordered in the crystal structure. In the MD simulations of gp41FP-TM-2H10 in a lipid bilayer (Figure 1—figure supplement 5), and in the MD simulation of the post-fusion conformation model in the lipid bilayer (Figure 4), most modeled FP residues (512 to 517) stay helical in the membrane environment. In the MD post-fusion model, only residue A512 frays in one protomer, and residues A512 and V513 unfold in the other two protomers. In the gp41FP-TM-2H10 simulation in the bilayer, only residue A512 unfolds in all three protomers, suggesting that the N-terminal region of FP (residues 513/514 to 517) may, indeed, also adopt a helical conformation within a lipid bilayer.

At any rate, as modeled in Figure 1.

Figure 1 shows the model of the crystal structure; it only features those residues that are accounted for by electron density.

Figure supplement 4A, the two FP helices modeled end at the middle of the membrane, in the aliphatic moiety. The charged N-termini for the two chains modeled will be very unstable there, so there is no wonder that the MD moves them to reach the phosphate headgroups of the lipids across the membrane.

The structure shown in this figure supplement (now Figure 1—figure supplement 5) is based on the crystal structure. Only missing FP and TM residues have been modeled. We added the figure to emphasize that the crystal conformation is, indeed, stable within a lipid bilayer during MD simulations, which revealed only small movements of TM and FP, including occasional lipid head groups moving up to contact form noncovalent interactions with the N-terminus of FP. We reason that local thinning of the membrane could be important for the fusion process, as reported previously for SNARE catalyzed fusion (Smirnova et al. PNAS, 116, 2019). Nevertheless, we have toned down our statement on FP orientation in the manuscript.

This model goes against the current paradigm that the fusion peptides only interact with the outer leaflet of the target membrane, and does not cross it.

The current models of FP-lipid bilayer interaction are entirely based on isolated FP peptide structures and membrane interaction studies with FP peptides. These studies indicate indeed lateral immersion of FP into one lipid bilayer, which may well constitute a first encounter of FP with the lipid environment. However, the crystal structure shows that FP adopts a continuous helix with FPPR and HR1 in protomer B, and forms a helical extension in protomer C. As suggested in the manuscript, protomer B likely constitutes the protomer closest to the final post-fusion conformation, which indicates that FP crosses one leaflet of the bilayer within the complete gp41 structure.

We toned down the statement, and included lateral FP interaction in the model shown in Figure 5.

It is questionable how the model has been docked into the membrane. Figure 1 Figure supplement 4B indeed suggests that the two FP helices traverse the membrane to interact with the lipid headgroups of the inner leaflet. Please soften the statements regarding the molecular dynamics simulations and model and emphasize that it they are highly speculative.

The structure was docked into a bilayer by using TM residues K683 and R707 defining the membrane boundaries.

Lines 233-266. Related to the previous point, the authors should compare by also docking the FP on the membrane not as in the symmetrized X-ray structure, which is derived from the protein having this flexible non-polar region within a detergent micelle, but with the FP anchored only on the outer leaflet, as it would initially insert into target membranes in an extended intermediate conformation. In this respect, the illustration of Figure 5D is highly mis-leading, as the peptides are not expected to insert into the target membrane in the proposed way, with the charged N-termini within the aliphatic moiety of the membrane.

We would like to point out that Figure 5D is a model and no complete structure of this proposed intermediate conformation of gp41 is available. It has been modeled based on a recent intermediate conformation of influenza HA (Benton et al., Nature 583, 2020), which also suggests a flexible linkage of the central coiled coil to FP. We have added the model proposed by the reviewers that suggests that FP first interacts laterally with the outer leaflet of the host cell membrane. FP may then only interact with each other in a later conformation such as the post-fusion conformation MD model presented here. Notably, the conformation of a triple stranded FP coiled coil is robust within a lipid bilayer upon MD simulation, thereby corroborating that FP can span one leaflet of the bilayer.

Figure 5 indeed shows that lipid head groups move to the center of the membrane in order to compensate for the charge at the N-termini, which is highly unlikely. This Figure should be adapted to reflect this.

The models of Figures 5E and F correspond to the crystal structure (with helical extensions of FP and TM) and Figure 5G shows the MD post fusion model. We have added the following sentence to the description on page 9, line 303:

“Notably, in this model based on the crystal structure, FP spans only one leaflet of the lipid bilayer facilitating local lipid head group interactions with the N-terminus of FP.”

Lines 220-222. Did the authors check that 2F5 does not bind the gp41FP-TM construct? It could select for a different conformation than does 2H10 and still bind.

We did not test 2F5 binding to gp41FP-TM. We agree with the reviewers that it could select a different extended conformation than 2H10.

Or does it have to bind its epitope exclusively when in the context of pre-fusion gp160?

The initial target of MPER bnAbs is still a matter of debate. Only structural information on 10E8 interaction with an “open” Env conformation is available. Other studies indicate that MPER bnAbs interact with an extended fusion-intermediate conformation that bridges viral and host cell membranes (Frey et al. Proc Natl Acad Sci USA 105, 2008). It was indeed surprising to detect high-affinity interactions with the gp41 conformation presented here.

Because this antibody was used in the fusion experiments, it should be possible to test its binding properties. And the reverse question: Do 10E8 or LN01 inhibit fusion in the system described in the previous paragraph? Since the peptide used should include their epitopes, they could also have a fusion blocking effect.

We have included the 10E8 Fab in the fusion assay as described above, showing its inhibitory effect.